EMBO
Molecular Medicine

# Targeting FKBP51 prevents stress-induced preterm birth

Ozlem Guzeloglu-Kayisli [1,3✉], Asli Ozmen [1,3], Busra Cetinkaya Un [1], Burak Un [1], Jacqueline Blas[1], Isabella Johnson[2], Andrea Thurman[2], Mark Walters[2], David Friend[2], Umit A Kayisli [1] & Charles J Lockwood[1✉]

## Abstract

Preterm birth (PTB) is a leading cause of perinatal morbidity and mortality, with maternal stress-related disorders, such as depression and anxiety, linked to idiopathic PTB (iPTB). At the maternal-fetal interface, decidualized stromal cells (DSCs) exclusively express the progesterone receptor (PR) and play pivotal roles in maintaining pregnancy and initiating labor. DSCs also express FKBP51, a protein that binds to and inhibits transcriptional activity of glucocorticoid and PR receptors and is associated with stress-related diseases. We previously found that iPTB specimens exhibit increased FKBP51 levels and enhanced FKBP51-PR interactions in DSC nuclei. Additionally, we demonstrated that *Fkbp5*-deficient mice have prolonged gestation and are resistant to stress-induced PTB, suggesting that FKBP51 contributes to iPTB pathogenesis. Since no FDA-approved therapy exists for PTB, we hypothesized that inhibiting FKBP51 could prevent iPTB. Our current results show that the endogenous prostaglandin D2 derivative 15dPGJ2 reduces FKBP51 levels and FKBP51-PR interactions in cultured cells. Maternal stress increases uterine expression of *Fkbp5*, *Oxtr*, and *Akr1c18*, leading to shortened gestation. However, treatment with 15dPGJ2 lowers uterine *Fkbp51*, *Oxtr*, and *Ptgs2* levels and prevents stress-induced PTB. Notably, co-treatment with 15dPGJ2 and either P4 or R5020 produced the most significant effects, highlighting the potential of 15dPGJ2 alone or in combination with progestins as a promising therapeutic strategy to prevent PTB.

**Keywords** 15dPGJ2; Maternal Stress; Preterm Birth; FKBP51; Progesterone Receptor
**Subject Categories** Development; Urogenital System

## Introduction

Preterm birth (PTB), defined as parturition prior to 37 completed weeks of gestation, is a leading cause of perinatal morbidity and mortality, affecting 10.5% of live births in the United States (https://www.marchofdimes.org/peristats/reports/united-states/prematurity-profile, 2023; Hamilton et al, 2024). The rate of PTB is higher among certain groups, with racial, ethnic, and socioeconomic disparities contributing to a rate of 14–15% among Black women (Dimes, 2023; https://www.cdc.gov/reproductivehealth/maternalinfanthealth/pretermbirth.htm, Accessed on 05-01-2021). PTB and low birth weight were responsible for ~17% of infant deaths before age 1, and premature infants are also at higher risks for chronic lung disease, cerebral palsy, and other long-term health issues (https://www.cdc.gov/reproductivehealth/maternalinfanthealth/pretermbirth.htm, Accessed on 05-01-2021; Svedenkrans et al, 2013). About 70% of PTBs are spontaneous, with 30% due to deteriorating maternal and/or fetal conditions, referred to as iatrogenic PTB, such as preeclampsia or fetal growth restriction (Goldenberg et al, 2008; Lockwood and Kuczynski, 2001). Over half of spontaneous PTBs prior to 32 weeks result from genital tract infections, abruptions, or multiple gestations, while the majority of cases occurring at or after 32 weeks are classified as idiopathic PTBs (iPTBs), for which causes remain largely unknown (Goldenberg et al, 2008; Lockwood and Kuczynski, 2001). Though maternal stress, including depression, anxiety, post-traumatic stress disorder, allostatic load, and chronic stress, as well as fetal stress caused by lower birth weight and/or abnormal placentation are strongly associated with iPTB (Lockwood and Kuczynski, 2001; Ott, 1993; Yonkers et al, 2012; Yonkers et al, 2014), the exact mechanisms underlying the stress-related iPTB remain to be elucidated.

The decidua is strategically located between the myometrium and placenta/fetal membranes and plays paradoxical roles to both maintain pregnancy and promote parturition. In the decidua, decidualized stromal cells (DSCs) are the predominant cell type and the only cells that express progesterone receptors (PRs) (Lockwood et al, 2010; Merlino et al, 2009; Vento-Tormo et al, 2018). In most mammals, a decline in maternal serum progesterone (P4) levels precedes labor, but in humans, high P4 levels are sustained until delivery (Mesiano et al, 2011; Welsh et al, 2014; Zakar and Hertelendy, 2007), supporting the hypothesis that functional P4 withdrawal induces labor. Thus, decreasing PR levels or activity, and/or increasing local P4 metabolism initiates human labor by inducing decidual inflammation, myometrial contractility, fetal membrane rupture, and facilitating cervical remodeling (Challis et al, 2000; Lockwood et al, 2010; Menon et al, 2016; Mesiano et al, 2011; Schatz et al, 2015).

[1]Department of Obstetrics & Gynecology, Morsani College of Medicine, University of South Florida, Tampa, FL, USA. [2]Dare Bioscience Inc., San Diego, CA, USA. [3]These authors contributed equally: Ozlem Guzeloglu-Kayisli, Asli Ozmen. ✉E-mail: ozlem2@usf.edu; cjlockwood@usf.edu

FK506-binding protein 51 (FKBP51), encoded by the *FKBP5* gene, has been linked to stress-related pathologies e.g. depression, anxiety, aging, inflammation and obesity (Binder, 2009; Binder et al, 2004). In the cytosol, FKBP51 exists as part of a protein complex, interacting with heat shock proteins, and the unliganded glucocorticoid receptor (GR) (Ratajczak et al, 2003; Sanchez, 2012) and/or PR. Binding of P4 to PR or cortisol to GR triggers a shift in the protein complex from FKBP51 to FKBP52, which promotes receptor dimerization, nuclear translocation and PR or GR-mediated transcriptional activity (Davies et al, 2002; Riggs et al, 2004; Tranguch et al, 2006). Stress triggers cortisol secretion from adrenal glands by inducing hypothalamic corticotropin-releasing hormone (CRH) and pituitary adrenocorticotropic hormone secretion (ACTH). Cortisol then activates GR-mediated transcription and inhibits further release of CRH and ACTH, resulting in negative endocrine feedback in the hypothalamic-pituitary-adrenal (HPA) axis. On the other hand, cortisol also generates negative autocrine feedback (Hubler and Scammell, 2004; Riggs et al, 2004) by increasing FKBP51 levels (Schatz et al, 2015), which in turn inhibit the binding of P4 to PR or cortisol to GR. Thus, elevated FKBP51 levels reduce cellular responses to P4 and glucocorticoids (Denny et al, 2005; Hubler et al, 2003; Hubler and Scammell, 2004). We previously found elevated FKBP51 levels in DSC nuclei from laboring vs. non-laboring term placentas (Schatz et al, 2015) and higher FKBP51 levels accompanied by enhanced nuclear FKBP51-PR interaction in DSCs from iPTB vs. gestational age (GA)-matched control preterm specimens (Guzeloglu-Kayisli et al, 2021). In our term DSC cultures, *FKPB5* overexpression reduced, while silencing enhanced, PR binding to specific DNA progesterone response elements (PREs) (Schatz et al, 2015). In line with the role of FKBP51 in labor, *Fkbp5* deficient (*Fkbp5*$^{-/-}$) mice, compared to wild-type (*Fkbp5*$^{+/+}$) mice, exhibit prolonged gestation and resistance to stress-induced PTB (Guzeloglu-Kayisli et al, 2021). This resistance is due to reduced uterine expression of: (1) aldo-keto reductase family 1, C18 (*Akr1c18; aka* 20α-hydroxysteroid dehydrogenase; 20α-HSD), an enzyme that inactivates P4 by converting it to 20-alpha-dihydro progesterone (20α-OHP4); and (2) oxytocin receptor (*Oxtr*), which induces myometrial contraction (Guzeloglu-Kayisli et al, 2021). Moreover, *Fkbp5*$^{-/-}$ mice display protection against stress-induced hormonal changes and depressive and anxiety-like behaviors (Schmidt et al, 2012). Conversely, *FKBP5* gene polymorphisms and/or epigenetic changes that increase FKBP51 expression are linked to depression and stress-related disorders (Appel et al, 2011; Binder, 2009; Binder et al, 2004), which are associated with spontaneous PTB (Latendresse and Ruiz, 2011; Mannisto et al, 2016). Collectively, these results highlight FKBP51 as a potential target to prevent or delay stress-associated PTBs.

Preventing PTBs remains challenging due to its multifactorial etiology. Until recently, Makena® (17-hydroxyprogesterone caproate), an injectable synthetic P4, was thought to reduce the risk of PTB recurrence in women with a history of spontaneous PTB and/or a short cervix (Nelson et al, 2021; Norman et al, 2018). However, Makena® was withdrawn from the market in 2023 due to its lack of efficacy (Makena, April 6 2023), leaving no approved therapeutic options for preventing PTB. Thus, we hypothesize that therapeutically targeting FKBP51 by reducing its levels and/or activity could provide a novel approach to preventing PTB. To test this

hypothesis, we evaluated the anti-parturition effects of 15-deoxy-Δ^12,14-prostaglandin J2 (15dPGJ2), an endogenous prostaglandin D$_2$ derivative (Scher and Pillinger, 2005; Sykes et al, 2014), used alone or in combination with either P4 or promegestone (R5020), a pure progestin agonist resistant to P4 metabolizing enzymes, in primary cultures of term DSCs and on an in vivo mouse model of maternal stress-induced PTB.

## Results

### 15dPGJ2 and fatty acid derivatives inhibit glucocorticoid-induced FKBP51 levels in term DSC cultures

Our initial screening of prior publication to identify agents that inhibit FKBP51 levels revealed a study indicating that 15dPGJ2 reduces FKBP51 levels induced by the dihydrotestosterone-bound androgen receptor (Kaikkonen et al, 2013). Further screening of the PubChem database for chemically and structurally similar compounds to 15dPGJ2 identified 9-nitro oleic acid (9 N) and 10-nitro oleic acid (10 N). Additionally, these three compounds are naturally produced in the body and function as endogenous PPAR-γ ligands (Borniquel et al, 2010; Kaikkonen et al, 2013). Thus, to test whether any of these agents inhibits basal and/or dexamethasone (Dex) induced FKBP51 levels in vitro, cultured term DSCs ($n = 3$) were treated with either vehicle control or $10^{-7}$ M Dex (Sigma-Aldrich, St-Louis, MO) or $10^{-5}$ M 15dPGJ2 (EMD Millipore, Temecula, CA), or 9N or 10N (Cayman, Ann Arbor, MI), either alone or in combination with Dex for 6 and 24 h.

Compared to control, Dex significantly increased *FKBP5* mRNA ($11.6 \pm 0.9$ vs. $1 \pm 0.02$) and protein ($0.67 \pm 0.06$ vs. $0.2 \pm 0.05$) levels (Fig. 1A,B). Treatment with 15dPGJ2, or 9N, or 10N alone did not alter endogenous *FKBP5* mRNA and protein levels. However, Dex-induced increases in *FKBP5* mRNA levels were inhibited ~2.2-fold by 15dPGJ2, ~1.5-fold by 9N, and 1.7-fold by 10N (Fig. 1A). Dex-induced increases in FKBP51 protein levels were also reduced ~2.1-fold by 15dPGJ2 and 1.5-fold by 9N, but not by 10N (Fig. 1B).

### 15dPGJ2 inhibits *FKBP5* levels and reduces FKBP51-PR binding

Our prior studies (Guzeloglu-Kayisli et al, 2021; Lockwood et al, 2010; Schatz et al, 2015) suggested that elevated FKBP51 levels contribute to both term and/or preterm birth by inducing functional P4 withdrawal through the inhibition of PR transcriptional activity. Thus, using these fatty acid derivatives to decrease FKBP51 levels and/or activity could represent a novel therapy for preventing PTB. Among these agents, we selected 15dPGJ2 for further experiments since it demonstrated the strongest inhibition of FKBP51 levels (Fig. 1A,B). Thus, we examined the effects of 15dPGJ2, either alone or in combination with P4, or R5020, on basal and glucocorticoid-induced *FKBP5* levels in term DSC cultures. Briefly, term DSCs ($n = 5$) were treated with $10^{-8}$ M estradiol (E$_2$, Sigma-Aldrich), $10^{-7}$ M P4 (Sigma-Aldrich), or $10^{-7}$ M R5020 (PerkinElmer, Waltham, MA) $\pm 10^{-5}$ M 15dPGJ2 (EMD Millipore) for 6 or 24 h for RNA and protein analyses, respectively. In a second set of cultures, term DSCs were pre-treated for 24 h with $10^{-7}$ M Dex to enhance basal FKBP51 levels, followed by the treatments described

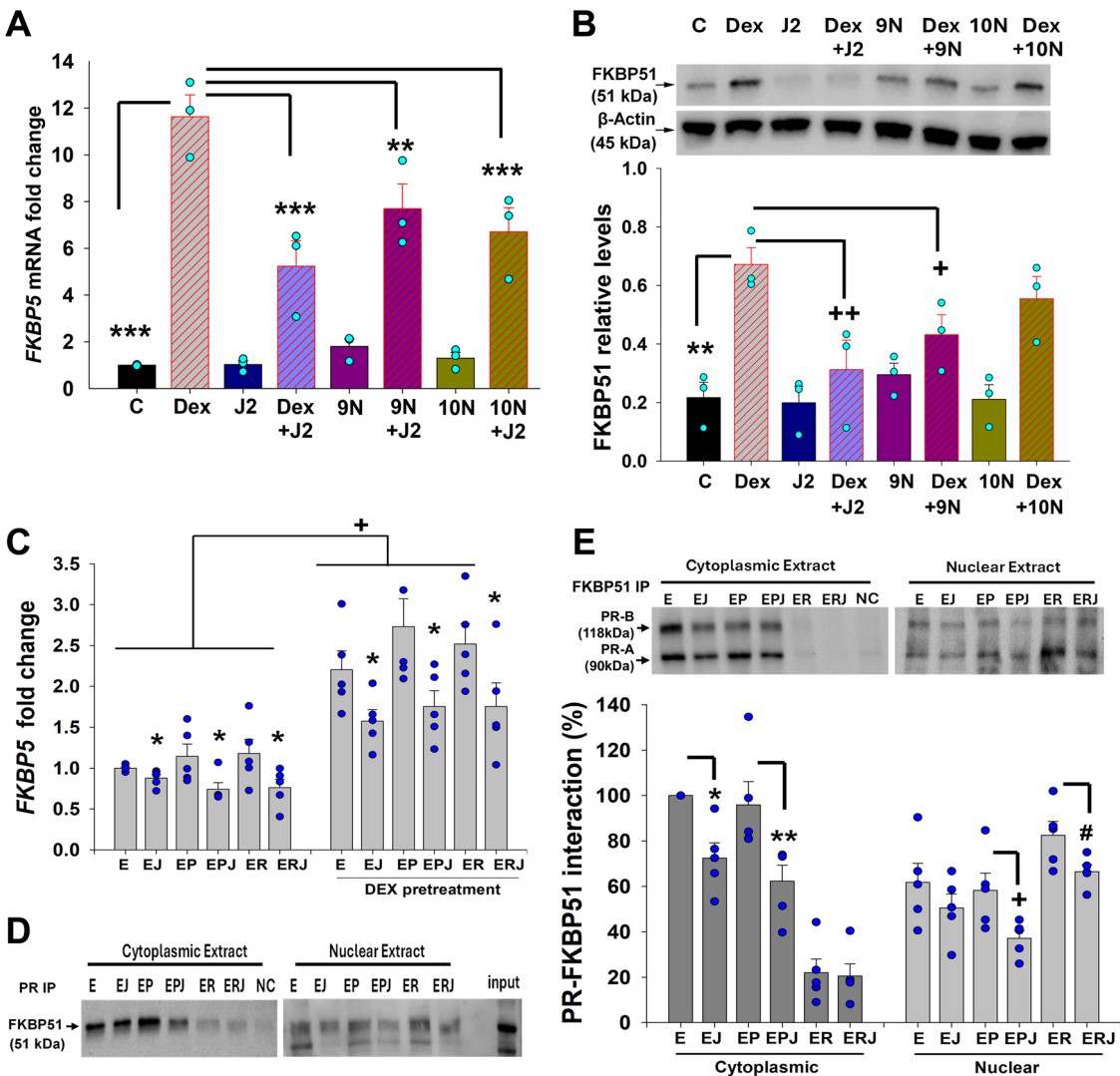

**Figure 1. 15dPGJ2 and fatty acid derivatives reduce dexamethasone-induced *FKBP5* mRNA and protein levels, and 15dPGJ2 reduced FKBP51-PR interaction.**

(A, B) *FKBP5* mRNA and protein levels were analyzed by qPCR and immunoblotting, respectively in term decidual stromal cells (DSCs) treated for 6 or 24 h with either control (C) or $10^{-7}$ M dexamethasone (Dex) or $10^{-5}$ M 15dPGJ2 (J2) or 9N or 10N-nitro oleic acids, alone or in combination with Dex. Data represents the fold change of *FKBP5* mRNA (A) and protein (B) levels, normalized to β-actin, presented as mean ± s.e.m.; $n = 3$ biological replicates. (A) ***$P < 0.001$ vs. Dex; **$P = 0.002$ vs. Dex. Statistical analysis was performed by One-Way ANOVA followed by Student–Newman–Keuls test. (B) **$P = 0.002$ vs. Dex; ++$P = 0.006$ vs. Dex; +$P = 0.04$ vs. Dex, by one-way ANOVA followed by Student–Newman–Keuls test. (C) Analysis of *FKBP5* levels by qPCR in term DSCs-pretreated with either vehicle or $10^{-7}$ M DEX for 24 h, then treated with $10^{-8}$ M estradiol (E) $\pm 10^{-5}$ M 15dPGJ2 (J) $\pm 10^{-7}$ M progesterone (P) or $\pm$R5020 (R) for 6 h. Relative expression of *FKBP5* was normalized to β-actin. Bars represent mean ± s.e.m.; $n = 5$ biological replicates. *$P < 0.05$ vs. corresponding E or EP or ER; +$P < 0.05$ DEX-pretreated vs. corresponding non-treated group. Statistical analysis was performed by one-way ANOVA with Student–Newman–Keuls test. (D, E) Immunoblotting analysis of FKBP51-PR interaction in protein complex immunoprecipitated using rabbit anti-PR antibody (D) or goat anti-FKBP51 (E) from T47D cultures treated with $10^{-8}$ M E $\pm 10^{-5}$ M J $\pm 10^{-7}$ M P or $\pm$ R for 24 h. Bars represent mean ± s.e.m.; $n = 5$ from four biological replicates. *$P = 0.01$ vs. E; **$P = 0.004$ vs. EP, with one-way ANOVA followed by Student–Newman–Keuls test. +$P = 0.03$ vs. EP; #$P = 0.04$ vs. ER by *t* test. NC: non-immune IgG control produced in primary antibody host used as negative control. IP: Immunoprecipitation. Source data are available online for this figure.

above. Analysis of qPCR results revealed that in term DSC cultures, 15dPGJ2 significantly reduced *FKBP5* levels by ~30% in each condition compared to corresponding controls (Fig. 1C). Additionally, we observed a significant increase in *FKBP5* levels (> 2.2-fold) in Dex-pretreated term DSCs treated with E₂ or E₂ + P4 or E₂ + R5020 vs. non-treated cultures (Fig. 1C). A slight increase in *FKBP5* levels was also observed in term DSCs treated with E₂ + P4

or E₂ + R5020 following 24 h Dex pre-treatment. Treatment with 15dPGJ2 significantly reduced Dex-induced *FKBP5* levels by ~30% in each condition (Fig. 1C).

Immunoprecipitation experiments were conducted using the T47D cell line, which exhibits strong PR-A/B expression, since term DSCs display reduced PR-A/B expression under culture conditions, resulting in weak immunoprecipitation signals for FKBP51-PR

interaction. The cultures were pre-treated for 24 h with $E_2$ or $E_2 + P4$ or $E_2 + R5020$, followed by co-incubation with $10^{-5}$ M 15dPGJ2 for 24 h to investigate the impact of 15dPGJ2 treatment on PR-FKBP51 interaction. Cytoplasmic and nuclear extracts were immunoprecipitated using either goat anti-FKBP51 (R&D, Minneapolis, MN) or rabbit anti-PR antibody (Cell Signaling, Danvers, MA). Immunoblotting analysis of immunoprecipitated complexes revealed the following: (1) FKBP51-PR interaction occurs in both cytoplasm and nucleus; (2) 15dPGJ2 in combination with $E_2 + P4$, significantly decreases PR-FKBP51 interaction in both the cytoplasm and nucleus; and (3) 15dPGJ2 in combination with $E_2 + R5020$, significantly reduced PR-FKBP51 interaction in the nucleus, but not in the cytoplasm, likely due to the more rapid translocation of PR to the nucleus from the cytoplasm and absent nuclear progestin metabolism (Fig. 1D,E). These results indicate that 15dPGJ2 reduces FKBP51 levels and lowers FKBP51-PR interaction, leading to enhanced P4-PR or P4-R5020 binding, which suggests enhanced PR transcriptional activity due to elevated levels of liganded PR in the nucleus.

## 15dPGJ2 inhibits levels of labor initiating molecules in cultured term DSCs

In contrast to pro-inflammatory effects of many prostaglandins, 15dPGJ2 exhibits strong anti-inflammatory properties (Lindstrom and Bennett, 2005a; Scher and Pillinger, 2005) via inhibiting the NF-κB, and AP-1 signaling pathways, which are key mediators of labor initiation (Lindstrom and Bennett, 2005b). Therefore, to provide in vitro evidence supporting the use of 15dPGJ2 in preventing PTB, primary term DSC cultures were tested to assess the inhibitory effects of 15dPGJ2 on key mediators of labor initiation. Analysis of qPCR results revealed that in term DSC cultures: (1) neither P4 nor R5020 altered expression of *IL1B* or *OXTR* mRNA levels (Fig. 2A,B); (2) 15dPGJ2 significantly inhibited both *IL1B* and *OXTR* mRNA expression (Fig. 2A,B); (3) Dex-pretreatment significantly reduced *IL1B* by twofold, but not *OXTR* levels, vs. non-treated term DSC cultures (Fig. 2A,B); and (4) 15dPGJ2 further significantly reduced *IL1B* mRNA levels by >5.5-fold in Dex-pretreated term DSCs (Fig. 2A), confirming that inhibition of FKBP51 by 15dPGJ2 enhances anti-inflammatory action of Dex-induced GR. Next, we assessed term DSC cultures treated with either vehicle or 0.1, 1, 2.5, 5 and 10 μM 15dPGJ2 for 6 h and found a concentration-dependent inhibition of *IL1B* or *OXTR* levels by 15dPGJ2, starting at 2.5 μM vs. control (Fig. 2C,D). Elevated levels of cyclooxygenase 2 (COX2) encoded by *PTGS2* gene, also contribute to term and preterm labor, by increasing levels of prostaglandin $F_{2\alpha}$ ($PGF_{2\alpha}$). Both $PGF_{2\alpha}$ and its analogs are clinically used as labor initiating agents (Luckas and Bricker, 2000). Thus, we measured *PTGS2* levels by qPCR and $PGF_{2\alpha}$ levels in collected condition media supernatant (CMS) obtained from term DSC cultures by ELISA. We found that $10^{-5}$ M 15dPGJ2 alone reduced *PTGS2* mRNA levels by 37%, and when combined with P4 or R5020, it reduces *PTGS2* levels by 54.8% and 49%, respectively (Fig. 2E), as well as secreted $PGF_{2\alpha}$ levels by ~60% (Fig. 2F) vs. controls. Collectively, our in vitro findings provide strong evidence that 15dPGJ2 significantly reduces FKBP51 levels, inhibits key mediators of labor initiation and exhibits strong anti-inflammatory effects in term DSC cultures.

## 15dPGJ2 does not affect P4 metabolism in term DSC cultures

In DSC cultures, P4 is quickly metabolized and returned to the basal levels after 24 h, indicating active intracellular P4 metabolism (Arici et al, 1999). To investigate whether 15dPGJ2 prevents conversion of P4 to its major inactive metabolite, 20α-OHP4, which normally increases during labor (DeTomaso et al, 2024; Nadeem et al, 2016; Paul et al, 2023; Williams et al, 2012) or other metabolites, CMS collected from 24 h term DSC cultures treated with $E_2$ or $E_2 + P4$ or $E_2 + R5020 \pm 1$ or 10 μM 15dPGJ2 ($n = 6$/group) were assessed by ultraperformance liquid chromatography-tandem mass spectrometry (UPLC/MS; Creative Proteomic, Shirley, NY). UPLC/MS results revealed that: (1) as expected, P4 levels or its metabolite 20α-OHP4 levels are significantly higher in $E_2 + P4$ group vs. $E_2$ or $E_2 + R5020$ group (Fig. 3A,B); (2) 1 or 10 μM 15dPGJ2 treatment does not alter either P4 levels or 20α-OHP4 levels in $E_2 + P4$ (Fig. 3A,B); and (3) 20α-OHP4 is the major metabolite (Fig. 3B) among estrogen, glucocorticoid, mineralocorticoid and androgen metabolites in CMS from term DSCs (Table EV1).

## 15dPGJ2 co-treatment with P4 or R5020 revealed an additive effect to block maternal stress induced PTB in a mouse model

To directly assess the role of 15dPGJ2, P4 and R5020 in preventing maternal stress induced PTB, and to test the hypothesis that combining 15dPGJ2 with progestins generates an additive impacts on PTB prevention, time mated pregnant mice were left either unrestrained (undisturbed control, NST) or subjected to maternal restraint-induced stress (ST), applied 3x/day for 1 h starting at embryonic day 8 (E8) through E18. Starting at E14 through E18, unrestrained mice ($n = 12$) received placebo injection while restraint pregnant mice were randomly allocated to receive injection of: 1) placebo ($n = 12$); (2) intraperitoneal (i.p.) 15dPGJ2 (20 μg/dam; $n = 13$); (3) subcutaneous (s.c.) P4 (0.2 mg/dam; $n = 13$); 4) i.p. 15dPGJ2 + s.c. P4 ($n = 13$); (5) s.c. R5020 (0.2 mg/dam single injection at E15; $n = 13$); and (6) i.p. 15dPGJ2 + s.c. R5020 ($n = 14$). Delivery timing was detected by monitoring mice from E18 through E22 every 4 h until completion of delivery and gestational length was calculated. As we reported previously (Guzeloglu-Kayisli et al, 2021), maternal restraint stressed mice displayed a significantly shortened gestational length, leading to PTB (mean ± s.e.m.: $18.7 \pm 0.02$ vs. NST mice $19.2 \pm 0.09$ days; Fig. 4A). Administration of 15dPGJ2 or P4 prevented stress-induced PTB by significantly prolonging gestational length in restrained mice ($19.1 \pm 0.07$ and $19.1 \pm 0.06$, respectively, vs. $18.7 \pm 0.02$; Fig. 4A). Notably, the combination of 15dPGJ2 with P4 ($19.6 \pm 0.1$) exhibited an additive effect, by further significantly extending gestational length compared to either 15dPGJ2 or P4 alone administration (Fig. 4A).

In control mice, 83.3% of term parturition occurred between E19 and E19.75, with only 8.3% experiencing PTB by being born prior to E19 or delayed labor with birth occurring at E19.75 or later (Fig. 4B). In contrast, 100% of the restraint stressed mice exhibited PTB (Fig. 4B). Among treatment groups, 69.2% of mice treated with 15dPGJ2 and 84.6% of those treated with P4 achieved term

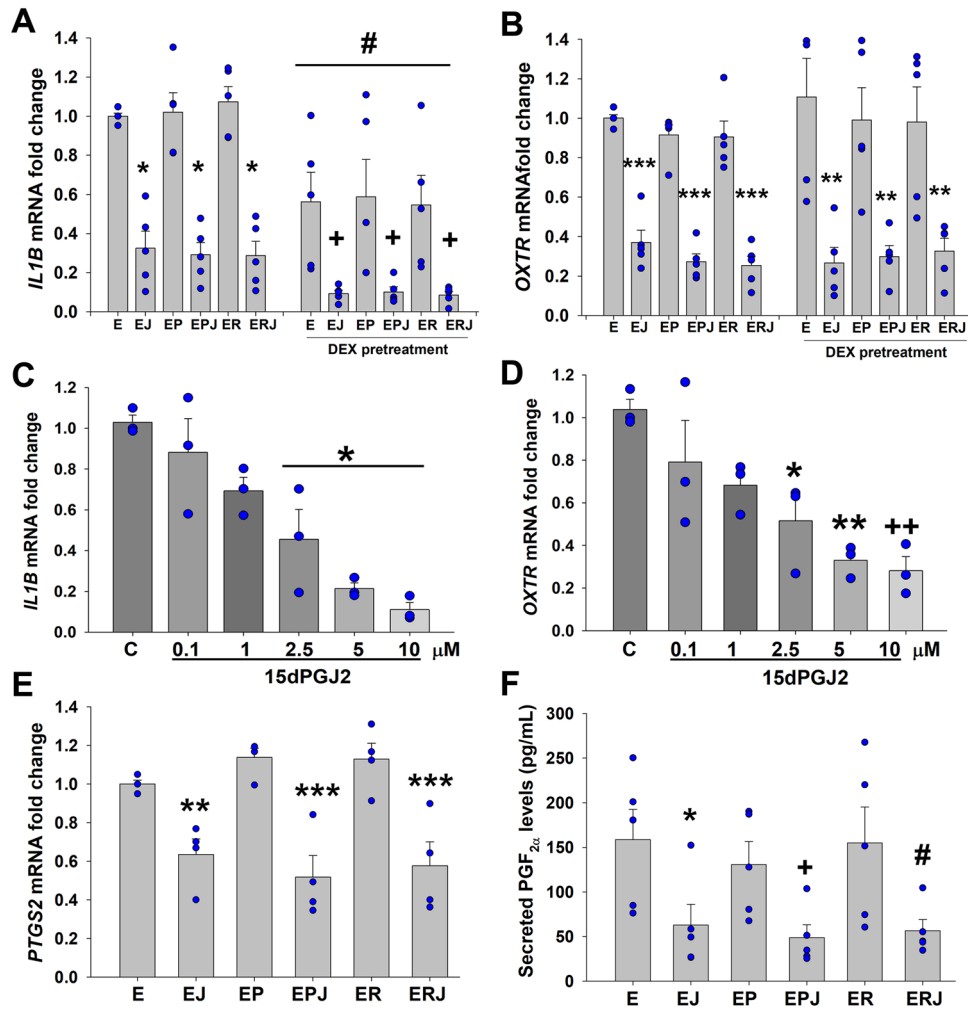

**Figure 2. 15dPGJ2 inhibits expression levels of genes, known to be key mediators of labor initiation.**

(A, B) Analysis of *IL1B* (A) and *OXTR* (B) levels by qPCR in term DSCs pretreated with either vehicle or $10^{-7}$ M dexamethasone (DEX) for 24 h, then treated with $10^{-8}$ M estradiol (E) $\pm 10^{-5}$ M 15dPGJ2 (J) $\pm 10^{-7}$ M progesterone (P), or $\pm$ R5020 (R) for 6 h. Bars represent mean $\pm$ s.e.m., $n = 5$ biological replicates. (A) *P < 0.05 vs. E, EP or ER; $^{+}P < 0.05$ vs. DEX-pretreated E, or EP or ER; $^{\#}P < 0.05$ DEX-pretreated vs. corresponding non-treated group. (B) ***P < 0.001 vs. E, EP or ER; **P = 0.003 or P = 0.007 or P = 0.002 vs. DEX-pretreated E, or EP or ER, respectively. Statistical analysis was performed by one-Way ANOVA followed by Student–Newman–Keuls test. (C, D) Concentration-dependent inhibition of *IL1B* (C) or *OXTR* (D) levels by 15dPGJ2. Bars represent mean $\pm$ s.e.m., $n = 3$ biological replicates; (C) *P < 0.05 vs. control; (D) *P = 0.02 or **P = 0.004 or $^{++}P = 0.003$ vs. control, by one-Way ANOVA with Student-Newman-Keuls test. (E) Analysis of *PTGS2* levels by qPCR in term DSCs treated with $10^{-8}$ M E $\pm 10^{-5}$ M J $\pm 10^{-7}$ M P, or $\pm$ R for 6 h. $n = 4$ biological replicates, **P = 0.008 vs. E; ***P < 0.001 vs. EP or ER, with one-way ANOVA followed by Student–Newman–Keuls test. (F) ELISA analysis of secreted $PGF_{2\alpha}$ levels in conditioned media supernatant obtained from DSC cultures. Data normalized to total protein concentration. $n = 5$ biological replicates; *P = 0.04 vs. E, $^{+}P = 0.02$ vs. EP by t test, and $^{\#}P = 0.03$ vs. ER, analyzed by Mann–Whitney U test. (A–E) Bars represent the fold change of expression levels, normalized to β-actin, presented as mean $\pm$ s.e.m, Source data are available online for this figure.

parturition, while 30.7% and 15.4%, respectively, experienced PTB. However, mice treated with 15dPGJ2 + P4 showed a 46% rate of term delivery and a 38.5% rate of delayed term labor (Fig. 4B). Moreover, administration of 0.2 mg/dam of R5020 from E14 to E18 prevented stress-induced PTB and delayed labor vs. restraint stressed mice (21.8 $\pm$ 0.09 vs. 18.7 $\pm$ 0.06; Fig. EV1A), resulting in no livebirths, with all pregnancies leading to either stillbirth (55.6% on E21.63) or dystocia (44.4% on E22, when the mice were euthanized, Fig. EV1B). Given the strong anti-labor inducing effects of R5020, likely due to its resistance to intracellular P4 metabolizing enzymes, we tested lower R5020 doses (0.1 and 0.05 mg/dam) *s.c.* daily from E14 to 18 and obtained similar adverse

pregnancy outcomes (stillbirth or dystocia). We then administrated 0.2 mg R5020 just once, either on E14 or E15 or E16. The results revealed that compared to restraint stressed mice (18.7 $\pm$ 0.06): (1) R5020 injection on E14 (18.7 $\pm$ 0.06, $n = 5$, Fig. EV1A) did not prevent stress-induced PTB, with 100% of mice having live births (Fig. EV1B); (2) R5020 injection on E15 (19.6 $\pm$ 0.3, $n = 13$; Figs. EV1A and 4A) significantly prolonged gestational length and reduced the stillbirth or dystocia rate to 23% (Figs. EV1B, 3 and 4C) R5020 injection on E16 (21.4 $\pm$ 0.4, $n = 5$) delayed labor (Fig. EV1A) but resulted in a very high stillbirth rate of 80% (Fig. EV1B). These findings highlight the strong progestogenic effect of R5020 and the critical role of timing its administration as

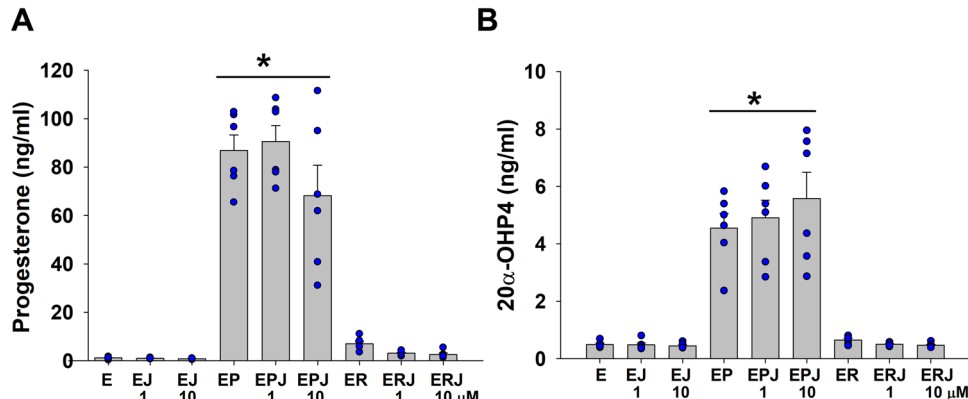

**Figure 3.    15dPGJ2 did not alter progesterone or its metabolite 20α-hydroxyprogesterone levels (20α-OHP4) in term decidual stromal cell cultures (DSCs).**

(A, B) Ultra-performance liquid chromatography-tandem mass spectrometry results showing progesterone (A) and 20α-OHP4 (B) levels in conditioned media supernatant obtained from term DSCs treated with $10^{-8}$ M estradiol (E) ± 1 or 10 μM 15dPGJ2 (J) ± $10^{-7}$ M progesterone (P) or ± R5020 (R) for 24 h. (A, B) Bars represent mean ± s.e.m.; $n = 6$ biological replicates. *$P < 0.05$ vs. any group, statistical analysis was performed by one-way ANOVA followed by Student–Newman–Keuls test. Source data are available online for this figure.

our studies optimized R5020 treatment to E15 to effectively prevent PTB. Moreover, gestational length was further prolonged in 15dPGJ2 + R5020 administrated mice (20.2 ± 0.3, $n = 14$) compared to administration of 15dPGJ2 or R5020 alone (Fig. 4A). In mice treated with 15dPGJ2 + R5020, 42.8% delivered at term, while 50% experienced delayed parturition (Fig. 4B). Stillbirth or dystocia were detected in 23% of the R5020 alone treated mice, while the rate was 43% in mice treated with both 15dPGJ2 and R5020. No such outcomes were observed in the other treatment groups (Fig. 4C).

## Administration of 15dPGJ2 or P4 or R5020 does not alter litter size

To evaluate the potential side effects of administrating 15dPGJ2, P4, or R5020 on litter size, we recorded the number of live and dead pups per litter immediately after completion of delivery. Our analysis revealed similar litter sizes (Fig. 4D) and comparable numbers of dead pups (Fig. EV1C) across control mice, and restraint stressed or 15dPGJ2 or P4 or R5020 treated mice. We also assessed fetal outcomes by recording the number of fetuses and the rate of fetal resorption at E18.25. We observed similar fetal numbers across all groups (Fig. EV1D). Additionally, the fetal resorption rates were comparable among the groups (Fig. EV1E), indicating that the treatments did not have a significant impact on fetal viability or early pregnancy loss. Importantly, we did not identify any malformed fetuses in any of the groups, further indicating that the alone or combined administration of 15dPGJ2, P4, or R5020 did not result in congenital or teratogenic effects. These findings indicate that these treatments do not cause significant adverse effects on fetal development or early pregnancy loss.

## Prenatal stress induced fetal growth restriction was reversed by 15dPGJ2 or P4 or R5020

Maternal stress is also associated with reduced birth weight and increases the risk for fetal growth restriction (Goyal et al, 2019).

Thus, to determine the effects of maternal restraint stress on birth weight, with or without administration of 15dPGJ2 or P4 or R5020, we weighed the pups immediately after delivery. Maternal stress significantly reduced birth weight in restrained vs. control mice (0.98 ± 0.01 vs. 1.32 ± 0.01 gram; Fig. 4E). However, birth weight was improved by administration of 15dPGJ2 (1.18 ± 0.01) or P4 (1.13 ± 0.01) or R5020 (1.14 ± 0.01) in restraint stressed mice (Fig. 4E). Moreover, pups from mice treated with 15dPGJ2 + P4 (1.21 ± 0.01) or 15dPGJ2 + R5020 (1.22 ± 0.01) had significantly higher birth weight compared to those from mice treated with P4 or R5020 alone (Fig. 4E). To assess sex-specific effects on birth weight, we analyzed female and male newborns separately. Female and male newborns exposed to maternal stress had birth weights which were 27.1% and 24.7% lower, respectively, compared to their control groups (Fig. 4F). Similar sex-specific trends were observed across other treatment groups, indicating that the protective effects of interventions such as 15dPGJ2 or P4 do not differ by sex (Fig. 4F).

We also measured pup weight at postnatal day 21 to explore if mice born with reduced birth weight experience catch-up postnatal growth. Compared to control mice, restrained female mice, but not restrained male mice, displayed significantly reduced body weight on postnatal day 21 (Fig. EV2). Prenatally stressed female or male mice exposed in utero to 15dPGJ2 or P4 or R5020 injections were not significantly different in their postnatal day 21 weight compared to control female or male mice (Fig. EV2).

## Tissues from offspring prenatally exposed to 15dPGJ2 or P4 or R5020 display normal histological structure

To assess potential side effects of 15dPGJ2, P4, or R5020 administration on offspring, we examined male and female pups in each group, all of which appeared phenotypically normal and exhibited no obvious growth and gross developmental anomalies. For more detailed analysis, pups ($n = 5$/each) were euthanized on postnatal day 21 and tissues including brain, heart, kidney, testis, ovary, and uterus were harvested and stained with H&E for

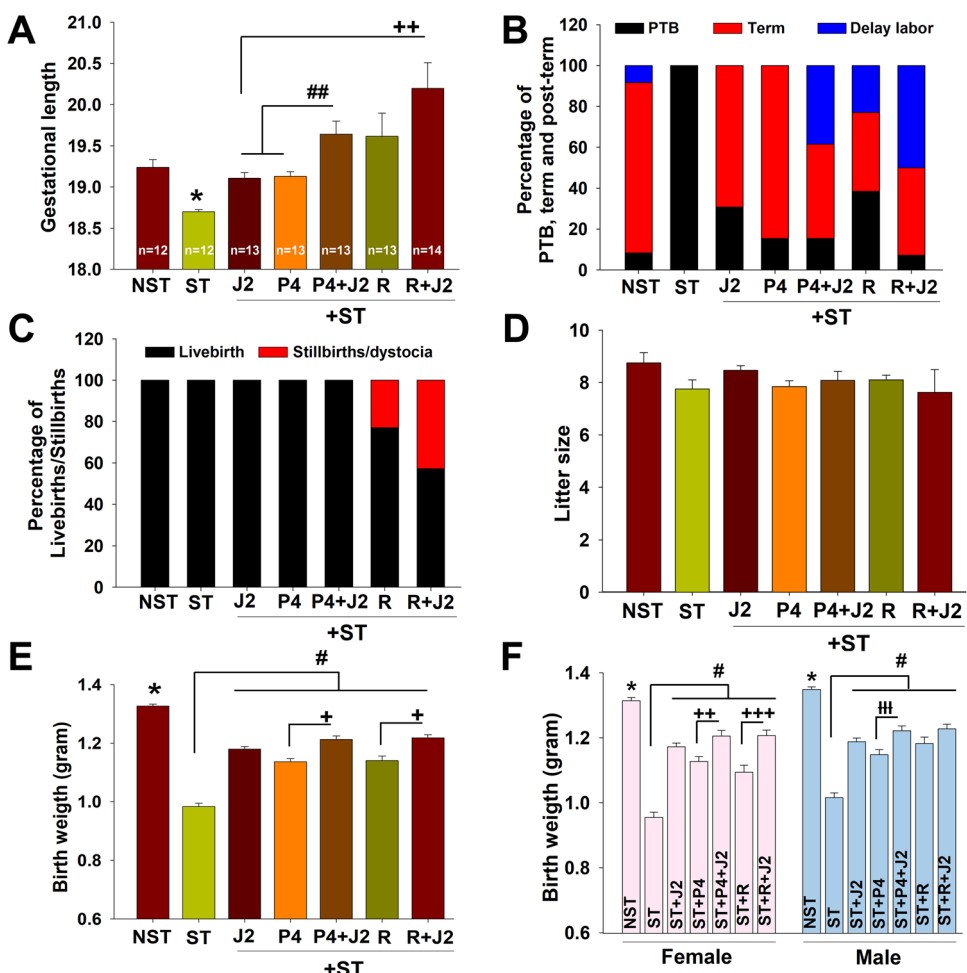

**Figure 4. 15dPGJ2 alone prevents maternal stress-induced PTB in mice, and its co-administration with either P4 or R5020 further extends gestation through additive effects.**

(A–C) Gestational length (A), percentage of preterm (PTB, <E19.0), term (E ≥ 19 to ≤19.75) and post-term ( > E19.75) birth (B), and percentage of live birth and stillbirths or dystocia (C) in time-pregnant wild-type mice left undisturbed state as control (NST, $n = 12$) or subjected to maternal restraint stress from E8 to E18 and received from E14 to 18 injection of placebo (ST, $n = 12$) ± 15dPGJ2 (J2, $n = 13$), or P4 (P4, $n = 13$) or P4 + 15dPGJ2 (P4 + J2, $n = 13$), or R5020 (R, $n = 13$) or R5020 + 15dPGJ2 (R + J2, $n = 14$). (A): *$P < 0.05$ vs. any group, with one-way ANOVA followed by Dunn's method; ##$P = 0.01$ vs. J2 or P4, ++$P = 0.003$ vs. J2, and $P = 0.08$ R vs. R + J2, analyzed by Mann–Whitney $U$ test. (D) No effect of maternal stress± 15dPGJ2 or P4 or R5020 alone or co-administrations on litter size obtained from NST ($n = 12$) or subjected to maternal restraint stress (ST, $n = 12$) ± J2 ($n = 13$), or P4 ($n = 13$) or P4 + J2 ($n = 13$), or R ($n = 10$) or R + J2 ($n = 8$). Mice exhibiting stillbirth/dystocia in R or R + J2 group are excluded. $P = 0.21$, with one-way ANOVA. (E, F) The effects of maternal stress± 15dPGJ2 or P4 or R5020 alone or co-administrations on birth weight all together (E) or according to gender of pups (F). (E) *$P < 0.05$ vs. any group; #$P < 0.05$ vs. ST; +$P < 0.05$ vs. ST + P4 or ST + R, analyzed with one-way ANOVA followed by Dunn's method. Total number of pups measured: NST $n = 84$, ST $n = 73$; J2 $n = 99$; P4 $n = 83$; P4 + J2 $n = 73$; R $n = 66$, and R + J2 $n = 53$. (F) *$P < 0.05$ vs. any group; #$P < 0.05$ vs. ST, with one-way ANOVA followed by Dunn's method. ++$P = 0.002$ vs. ST + P4, +++$P = 0.001$ vs. ST + R, with Mann–Whitney $U$ test, and ‡‡$P = 0.001$ vs. ST + P4, $P = 0.08$ ST + R + J2 vs. ST + P4, analyzed by $t$ test. Total number of female pups measured: NST $n = 53$, ST $n = 38$; J2 $n = 48$; P4 $n = 46$; J2 + P4 $n = 35$; R $n = 31$; and J2 + R $n = 25$. Total number of male pups measured: NST $n = 31$, ST $n = 35$; J2 $n = 51$; P4 $n = 37$; J2 + P4 $n = 38$; R $n = 35$; and J2 + R $n = 28$. Note that R5020 was only single injection given on E15 (A–F). Bars represent mean ± s.e.m. (A, D, E, F). Source data are available online for this figure.

histopathological evaluation. All tissues were examined under the light microcopy for: (1) cellular irregularities, necrotic areas and/or fibrotic depositions; (2) presence of leucocyte infiltration; or (3) hemorrhagic areas. Offspring tissues from dams exposed to maternal stress ± 15dPGJ2 or P4 or R5020 injections displayed similar cellular and extracellular structures with no obvious necrotic, fibrotic, hemorrhagic and/or leukocyte infiltration areas compared to offspring tissues obtained from control dams (Fig. EV3).

## Prenatal 15dPGJ2 exposure does not impact the reproductive outcomes of offspring

P4 use during pregnancy is not reported to affect subsequent fertility of offspring (https://www.drugs.com/pregnancy/progesterone.html). Since the potential side effects of 15dPGJ2 on fertility are unknown, we conducted a continuous mating study using: (1) sexually mature restraint stressed female as a control ($n = 10$) mated with control male mice; (2) stressed and prenatally

15dPGJ2 exposed female mice ($n = 7$) mated with control male mice; or (3) control female mice ($n = 9$) mated with prenatally 15dPGJ2 exposed male mice. Cages were monitored weekly, and the time to delivery after mating, along with the number of pups born, were recorded. After 6 weeks of mating, both 15dPGJ2 exposed male and female mice exhibited normal fertility and similar number of live births compared to control female mated with male mice (Table 1).

## 15dPGJ2 or P4 co-administration enhances serum P4 levels in mice

Serum P4 levels were measured at E18.25 in all groups by ELISA to investigate effects of 15dPGJ2, P4 or R5020 on luteal regression, which normally precedes labor in mice accompanied by a reduction in serum P4 levels (Sugimoto et al, 1997). We previously reported similar serum P4 levels between E16 to E18 in control and restrained mice, but a dramatic decrease in serum P4 levels in restrained mice on E18.25 (Guzeloglu-Kayisli et al, 2021). The current study also detected a similar dramatic decline in serum P4 levels at E18.25 in restraint stressed mice vs. control ($4.5 \pm 0.5$ vs. $10.5 \pm 4.05$ ng/ml; Fig. 5A). However, compared to restraint

stressed mice, serum P4 levels (Fig. 5A): (1) were higher in 15dPGJ2 or P4 alone injected mice ($12.6 \pm 4.1$, or $20.3 \pm 3.6$, respectively); (2) further increases occurred in 15dPGJ2 + P4 co-injected mice ($36.3 \pm 6.2$), supporting the additive effect of this combination on gestational length (Fig. 5A); and (3) were unaltered in R5020 alone or 15dPGJ2 + R5020 injected mice ($7.5 \pm 0.7$ or $6.5 \pm 0.9$, respectively; Fig. 5A). In our previous study, we showed that restraint stress exposed mice displayed significantly higher corticosterone levels vs. control mice at E16, 17 and 18. However, corticosterone levels in restraint stressed mice declined at E18.25 by reaching similar corticosterone levels seen in control mice (Guzeloglu-Kayisli et al, 2021). Consistent with this observation, the current study found that serum corticosterone levels were similar in restraint stressed mice with or without 15dPGJ2 or 15dPGJ2 + P4 injection versus control mice at E18.25 (Fig. 5B). However, serum corticosterone levels were marginally lower in P4 or R5020 or 15dPGJ2 + R5020 administered restraint stressed mice vs. restraint stressed mice (Fig. 5B).

## Maternal stress induced uterine *Fkbp5*, *Oxtr* and *Akr1c18* levels are reduced by 15dPGJ2 + P4 or 15dPGJ2+R5020

Uterine tissues collected at E18.25 were analyzed by qPCR for levels of key mediators of labor initiation including *Fkbp5*, *Oxtr*, *Akr1c18*, *Il1b* and *Ptgs2* to investigate the effects of 15dPGJ2 alone or in combination with P4 or R2050. Consistent with our previous observation in human decidua from iPTB specimens, we reported that uteri of unrestrained *Fkbp5*[+/+] mice displayed a significant increase in *Fkbp5* levels at E19 vs. E17.25 or E18.25, and maternal stress significantly elevated *Fkbp5* levels in *Fkbp5*[+/+] mice at both E17.25 and E18.25 compared with unrestrained *Fkbp5*[+/+] mice (Guzeloglu-Kayisli et al, 2021). In the current study, at E18.25, compared to control mice ($1.34 \pm 0.41$), maternal restraint stressed mice displayed significantly elevated uterine *Fkbp5* levels ($3.45 \pm 0.79$), which were significantly inhibited by 15dPGJ2 alone ($0.58 \pm 0.14$) or P4 ($0.8 \pm 0.17$) or R5020 ($1.15 \pm 0.33$) combinations (Fig. 6A). Similar to our previous study (Guzeloglu-Kayisli et al, 2021), *Oxtr* and *Akr1c18* levels are significantly enhanced in the

**Table 1. Fertility rates and litter sizes are similar in female and male mice prenatally exposed to 15dPGJ2 compared to control female and male mice.**

| | Number, *n* | Pups per litter mean ± s.e.m. | Days to delivery mean ± s.e.m. |
|---|---|---|---|
| Control female × control male (control group) | 10 | 9.1 ± 0.4 | 25.1 ± 1.2 |
| 15dPGJ2 exposed female x control male | 7 | 8.4 ± 0.3 P = 0.3 | 21.7 ± 0.9 P = 0.06 |
| Control female × 15dPGJ2 exposed male | 9 | 9.1 ± 0.4 P = 0.9 | 23.0 ± 1.8 P = 0.1 |

Fertility rates of female and male offspring mice, prenatally exposed to restraint stress from E8 to 18 and received 15dPGJ2 administration from E14 to E18 were assessed by mating restraint stressed male and female mice (as control) for 6 weeks and day to delivery as well as resultant litter sizes were recorded. Statistical analysis was performed by *t* test versus control group.

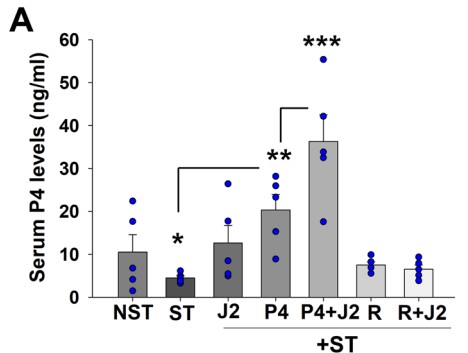
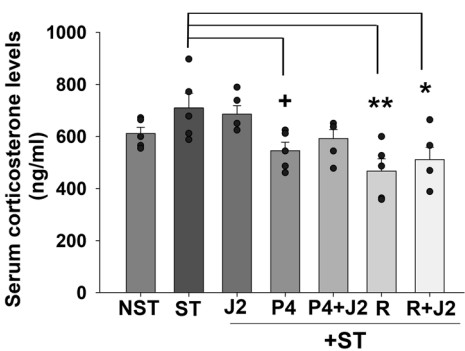

**Figure 5. 15dPGJ2 alone or co-administration with P4 elevates serum P4 levels.**

(A, B) Serum levels of P4 (A) and corticosterone (B) in time-pregnant wild-type mice left undisturbed state as control (NST) or subjected to maternal restraint stress between E8 and E18 and received from E14 to 18 injections of placebo (ST) ± 15dPGJ2 (J2), or P4 (P4) or P4 + 15dPGJ2 (P4 + J2), or R5020 (R) or R5020 + 15dPGJ2 (R + J2). (A) ***$P < 0.001$ vs. any group; **$P = 0.003$ vs. P4; *$P = 0.03$ vs. ST, with one-way ANOVA followed by Student–Newman–Keuls Method. (B) **$P = 0.004$, *$P = 0.02$ vs. ST, with one-way ANOVA followed by Student–Newman–Keuls Method, and +$P = 0.03$ vs. ST analyzed by *t* test. (A, B) Bars represent mean ± s.e.m. $n = 5$ biological replicates. R5020 was a single injection given on E15. Source data are available online for this figure.

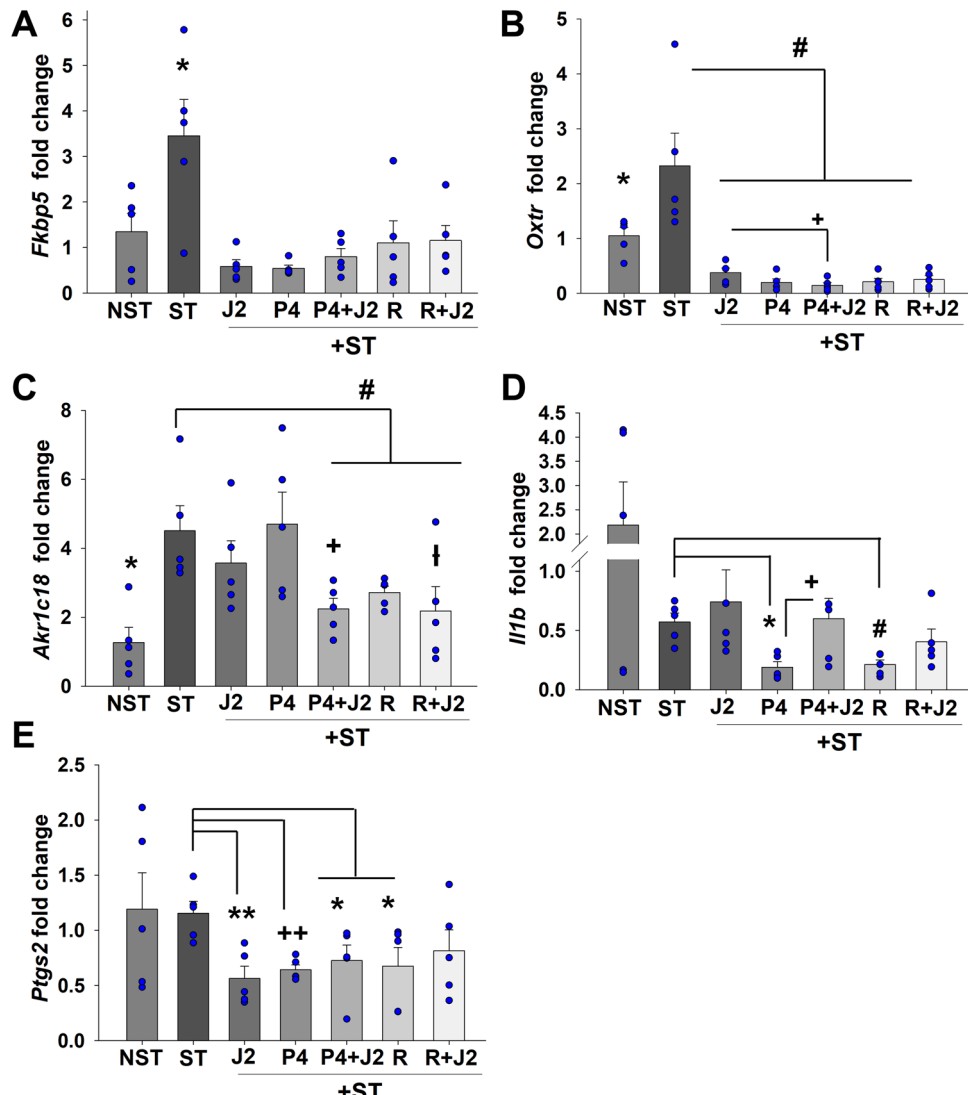

**Figure 6. 15dPGJ2 alone or co-administration with P4 or R5020 inhibits uterine expressions of *labor initiation mediators Fkbp5, Oxtr, Akr1c18, Il1b* and *Ptgs2* by in mice at E18.25.**

(A–E) In situ levels of *Fkbp5* (A), *Oxtr* (B), *Akr1c18* (C), *Il1b* (D) and *Ptgs2* (E) in time-pregnant mice left undisturbed state as control (NST) or subjected to maternal restraint stress between E8 and E18 and received from E14 to 18 injection of placebo (ST)± 15dPGJ2 (J2), or P4 (P4) or P4 + 15dPGJ2 (P4 + J2), or R5020 (R) or R5020 + 15dPGJ2 (R + J2). (A) *$P < 0.05$ vs. any group, with one-way ANOVA followed by Student–Newman–Keuls Method. (B) *$P < 0.05$ vs. any group; and #$P < 0.05$ vs. ST, with one-way ANOVA followed by Student–Newman–Keuls Method. +$P = 0.04$ vs. J2, analyzed by t test. (C) *$P < 0.05$ vs. any group; #$P < 0.05$ vs. ST; +$P < 0.05$ vs. J2 or P4; and †$P < 0.05$ vs. J2, with one-way ANOVA followed by Student–Newman–Keuls Method. (D) *$P = 0.002$ or #$P = 0.003$ vs. ST; and +$P = 0.045$ vs. P4, analyzed by t test. (E) **$P = 0.005$ or ++$P = 0.002$ or *$P = 0.04$ vs. ST, performed by t test. (A–E) Bars represent mean ± s.e.m. $n = 5$ biological replicates. R5020 was a single injection given on E15. Source data are available online for this figure.

uteri of restraint stressed mice vs. control mice (2.32 ± 0.59 vs. 1.05 ± 0.14 and 4.50 ± 0.72 vs. 1.26 ± 0.43; respectively, Fig. 6B,C). 15dPGJ2, alone or in combination with P4 or R5020 significantly inhibited *Oxtr* compared to restraint stressed or control mice (Fig. 6B). Mice treated with 15dPGJ2 + P4 displayed even greater *Oxtr* inhibition than those treated with 15dPGJ2 or P4 alone (Fig. 6B). 15dPGJ2 + P4 or 15dPGJ2 + R5020 injected mice showed significantly reduced *Akr1c18* levels vs. restraint stressed mice or 15dPGJ2 alone or P4 alone injected mice (Fig. 6C). *Uterine Il1b* levels were not altered significantly in restraint stressed vs. control mice. However, P4 or R5020 administration significantly

suppressed uterine *Il1b* levels vs. restraint stressed mice (Fig. 6D). Moreover, uterine *Ptgs2* levels were similar between restraint stressed and control mice whereas 15dPGJ2 alone or co-treatment with P4 or R5020 significantly suppressed uterine *Ptgs2* levels vs. restraint stressed mice (Fig. 6E). Inhibition of *Akr1c18* by 15PGJ2, when combined with P4 or R5020 prolongs the local P4 response due to reducing its metabolism of P4 to 20α-OHP4 in the uterus. This, in turn, further extends gestational length compared to the administration of either treatment alone, supporting the use of 15dPGJ2 + P4 as the optimal therapeutic approach in prevention of PTB.

# Discussion

The U.S. rate of PTB has remained consistently high over the past decade, highlighting the ongoing challenge of this public health issue (https://www.cdc.gov/reproductivehealth/maternalinfanthealth/pretermbirth.htm, Accessed on 05-01-2021; https://www.marchofdimes.org/peristats/reports/united-states/prematurity-profile, Accessed on 02-14-2023). Each year, PTB not only affects thousands of families but also imposes an economic burden exceeding $25 billion (Waitzman et al, 2021). Despite this, drug development in obstetrics, particularly for PTB, remains limited due to regulatory concerns involving pregnant patients, high development costs, and a limited market opportunity constrained by patent life. The complex etiology of PTB, including cervical insufficiency, preterm prelabor rupture of membranes, infections, decidual hemorrhage, and maternal and fetal stress require tailored therapeutic approaches for each condition to safely and effectively prolong pregnancy (Lilliecreutz et al, 2016). Given the absence of FDA-approved treatments for PTB, over 30,000 babies will be born preterm each month in the U.S. (https://www.marchofdimes.org/peristats/reports/united-states/prematurity-profile, Accessed on 02-14-2023), emphasizing the urgent need for innovative therapies that target specific causes of PTB. Such advancements could improve neonatal outcomes, long-term health for both mothers and children, and public health overall. Natural P4 (vaginal) or 17-hydroxyprogesterone caproate (intramuscular) supplementations have been used to prevent recurrent PTB. However, recent large randomized trials have failed to confirm their efficacy (Blackwell et al, 2020; Norman et al, 2016) while vaginal P4 is still indicated for women without a prior PTB who are incidentally found to have a short cervix (Hassan et al, 2011). Thus, clinicians face the challenge of questionable treatment options in the absence of FDA-approved PTB treatment. Although the exact causes of iPTB remain unknown, maternal stress, which can stem from conditions such as depression, anxiety, psychosocial stress or post-traumatic stress disorders, as well as fetal stress originating from abnormal placentation are strongly linked to iPTBs (Kovo et al, 2011; Latendresse, 2009; Lockwood and Kuczynski, 2001; Ott, 1993; Yonkers et al, 2012; Yonkers et al, 2014). Recent studies in humans and mice suggest that the main molecular mechanism behind the failure of this therapy is functional P4 withdrawal, which occurs independently of systemic P4 levels (Merlino et al, 2007; Mesiano et al, 2002; Thijssen, 2005). These studies indicate that disruption of P4-PR interaction in decidual, myometrial and cervical cells causes functional P4 withdrawal in both term and preterm labor, either by inhibiting PR action or levels and/or by enhancing local P4 metabolism (Challis et al, 2000; DeTomaso et al, 2024; Menon et al, 2016; Mesiano et al, 2011). Our prior findings uncovered two distinct mechanisms of functional P4 withdrawal contributing to PTB: (1) significantly reduced decidual PR levels in pregnancies complicated by chorioamnionitis or abruption, major causes of early spontaneous PTBs (Guzeloglu-Kayisli et al, 2015; Lockwood et al, 2012); and (2) increased decidual FKBP51 expression and enhanced FKBP51-PR interaction (Lockwood et al, 2010; Schatz et al, 2015) in pregnancies complicated by idiopathic and stress-associated PTBs, which accounts for the majority of late PTBs (Guzeloglu-Kayisli et al, 2021). Supporting the latter mechanism, we found that FKBP5-overexpression in decidual cell cultures elevates FKBP51-PR interaction and reduces PR binding to PRE, to block PR transcriptional activity, thereby causing functional P4 withdrawal (Schatz et al, 2015). Collectively, these data validate decidual FKBP51 as a key mediator of stress-induced PTB, leading us to screen therapeutic agents that inhibit FKBP51 activity and/or levels in decidual cells.

Pharmacological inhibition of FKBP51 activity for the prevention of stress-related disorders was targeted by SAFit1 and SAFit2, which provided initial proof-of-concept in animal models (Gaali et al, 2015). However, they are not fully characterized for preclinical studies due to limitations such as a poor physicochemical profile, unfavorable pharmacokinetics, and a higher molecular weight (Buffa et al, 2024). In the current study, our initial literature search identified 15dPGJ2 as inhibitor of dihydrotestosterone-induced FKBP51 levels (Kaikkonen et al, 2013). Further screening of the PubChem database for chemically and structurally similar agents to 15dPGJ2 revealed 9N and 10N as potential inhibitor of FKBP51 levels (Borniquel et al, 2010; Kaikkonen et al, 2013). Our subsequent in vitro studies confirmed 15dPGJ2, 9N, and 10N as the significant inhibitors of glucocorticoid induced FKBP51 levels in DSC cultures (Fig. 1), which strongly support their therapeutic potential in preventing PTB. The strongest inhibition of FKBP51 levels among these three agents was by 15 dPGJ2 which led us to select it for further investigation of its in vivo therapeutic impact on stress-induced PTB in our previously established mouse model (Guzeloglu-Kayisli et al, 2021). 15dPGJ2 has been extensively explored in tumors, inflammation, oxidative stress, fibrosis, vascular remodeling, and lipid metabolism (Behl et al, 2016; Bie et al, 2018; Scher and Pillinger, 2005). Moreover, 15dPGJ2 activates peroxisome proliferator-activated receptor γ (PPARγ), and inhibits NF-κB and AP-1 signaling, key mediators of labor initiation (Forman et al, 1995; Lindstrom and Bennett, 2005a; Scher and Pillinger, 2005; Sykes et al, 2014). However, to our knowledge, this is the first study demonstrating that 15dPGJ2 inhibits both endogenous and Dex-induced *FKBP5* levels as well as nuclear PR-FKBP51 interactions, thereby enhancing PR activity. As a result, increased PR activity suppresses expression of key mediators of labor initiation and inflammation, e.g., *OXTR and IL1B*, respectively (Fig. 7).

The inhibitory effect of FKBP51 on PR or GR mediated transcriptional activity (Smith, 2004) has been demonstrated in previous studies. Specifically, (1) overexpression of human or squirrel monkey *FKBP5* requires a 3- or 11-fold, respectively, higher P4 $EC_{50}$ for PR activation in HepG2 cells, respectively (Hubler et al, 2003); and (2) FKBP51 reduces cortisol induced GR activity by 2- or 12-fold, respectively in COS7 cells (Denny et al, 2005). These findings indicate that increased FKBP51 levels reduce P4 and glucocorticoid responses (Denny et al, 2005; Hubler et al, 2003; Hubler and Scammell, 2004). Our previous study also supports these results, showing that FKPB51 overexpression inhibits and silencing augments PR-DNA binding in cultured term DSCs (Schatz et al, 2015). Additionally, our recent study found that human decidual cells from iPTBs exhibit significantly higher in-situ FKBP51 levels and nuclear FKBP51-PR binding compared to GA-matched controls (Guzeloglu-Kayisli et al, 2021). Additionally, 15dPGJ2 exhibits anti-inflammatory properties by inhibiting *PTGS2* (aka *COX-2*) and *IL1B* levels in term DSCs. These anti-inflammatory effects of 15dPGJ2 were also observed in IL-1β induced human amniocyte and myocyte cultures, where 15dPGJ2 inhibited NF-κB activity in a manner independent of PPAR receptors (Lindstrom and Bennett, 2005a). Similarly, 30 μm 15d-PGJ2 reduced TNFα-stimulated *PTGS2* and $PGE_2$ levels in amniocytes via a PPARγ-independent mechanism (Ackerman et al, 2005) and inhibited LPS-stimulated IL-6, IL8 and TNFα

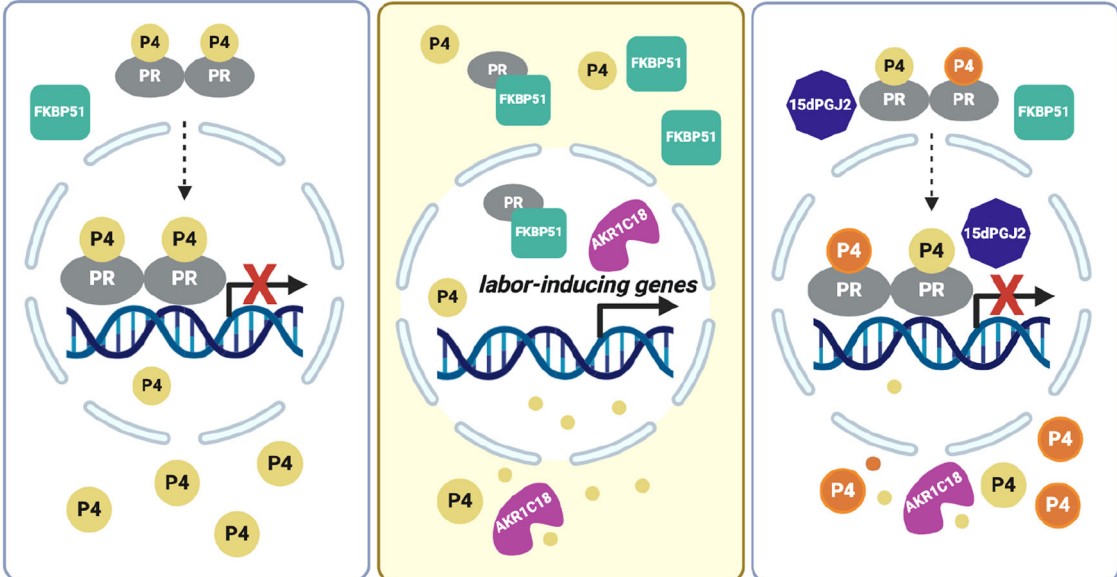

**Figure 7. FKBP51 mediates stress-induced PTB. Model illustrating the proteins involved in stress-induced PTB.**

The progesterone receptor-progesterone complex (PR-P4) promotes pregnancy maintenance by repressing labor-inducing genes (Left). Stress induces FKBP51 overexpression (green), leading to FKBP51 binding to PR (PR-FKBP51) and local P4 metabolism by ARK1C18 (magenta), collectively resulting in PTB due to the functional P4 withdrawal (Middle). Therapeutic 15dPGJ2/P4 strategy for the prevention of stress-induced PTB: 15dPGJ2 (blue) delivery suppresses FKBP51 levels, thereby restoring PR-P4 activity in decidual cells preventing stress-induced PTB (Right). P4 supplementation enhances PR activity as elevated FKBP51 levels necessitate higher nuclear P4 levels. Thus, 15dPGJ2/P4 co-treatment is more effective to reduce both FKBP51 and AKR1C18 levels. Image created in BioRender.

levels in amnion, choriodecidual and placental explants (Lappas et al, 2002). Given 15dPGJ2's potent anti-inflammatory properties, it may also be beneficial in reducing early-stage inflammation in infection-related PTB, particularly in situations where the mother and fetus are not at immediate risk.

Our prior studies showed that *Fkbp5*⁻/⁻ mice display longer gestations than *Fkbp5*⁺/⁺ and restraint stress induces PTB in *Fkbp5*⁺/⁺ mice, whereas *Fkbp5*⁻/⁻ mice are resistant to stress-induced PTB (Guzeloglu-Kayisli et al, 2021). These findings suggest that FKBP51 plays a crucial role on gestational length and/or initiation of parturition. 15dPGJ2 has been shown to delay LPS- or thrombin-induced PTB and improve pup survival in mice (Chigusa et al, 2016; Pirianov et al, 2009). However, to our knowledge, the current study is the first to demonstrate the 15dPGJ2 efficacy to prevent stress-induced PTB together with its safety and efficacy when used alone or in combination with P4 in a preclinical model of stress-induced PTB in mice. Significantly greater extension of gestation length by the 15dPGJ2 and P4 combinations compared to their alone administration in restraint stressed mice indicates that the combination has an additive effect and carry stronger therapeutic potential for preventing stress-associated iPTBs.

Compared to P4, the administration of R5020 from E14 to E18, until E22, substantially extended gestation further causing stillbirth/dystocia. This effect is likely due to R5020's resistance to P4-metabolizing enzymes, thereby maintaining its potent progestogenic effects to sustain pregnancy (Shynlova et al, 2022). However, a single injection of R5020 on E15 significantly prolonged gestation, but led to term parturition at E19.6, which reduced stillbirth/dystocia. A similar effect of R5020 (0.2 mg) was also reported in mouse model of systemic inflammation induced PTB (Shynlova et al, 2022), while 1 mg of P4

administration partially inhibited LPS-induced parturition without extending gestational length (Shynlova et al, 2022). This is consistent with a prior study (Hirsch and Muhle, 2002) showing that P4 supplementation did not fully prevent E.coli-induced PTB in mice. Moreover, the current study also revealed that combining a single dose of R5020 on E15 with 15dPGJ2 given from E14 to E18 further prolonged gestational length, similar to the additive effect observed with 15dPGJ2 and P4 co-administration.

Maternal stress can also cause fetal growth restriction, significantly reducing birth weight by 25.7% in pups from maternal restraint-stressed mice compared to control mice as previously reported (Vaughan et al, 2012). 15dPGJ2 treatment, either alone or in combination with P4 or R5020, prevented this growth restriction, reducing the birth weight deficit to 10.6%, 8.3% and 7.6%, respectively, in restraint stressed versus control mice. Neither 15dPGJ2 alone nor in combination with P4 caused obvious side effects, as there was no change in the viable pup percentages, litter size or evidence of gross anomalies or histological abnormalities. However, mice treated with 15dPGJ2 + R2050 showed an increased stillbirth and dystocia due to prolonged gestation. Comparable fertility and live birth rates were observed in both 15dPGJ2 in utero exposed male and female mice compared with control mice indicating the safety of 15dPGJ2 on subsequent reproductive outcomes. P4 is considered safe for use during pregnancy and is classified as Category B under the U.S. FDA Pregnancy risk assessment (Drugs.com, 2023; Leek JCAH, 2024). Based on these findings, we demonstrated that the combination of 15dPG2 with P4 is the most efficacious and safe therapeutic approach for preventing stress-induced PTB, improving birth weight, and maintaining litter size in our preclinical animal model.

Adequate levels of P4 are necessary to maintain pregnancy in all mammals. Unlike humans, P4 levels in mice decrease due to luteal regression to initiate labor (Sugimoto et al, 1997). We previously observed that restrained $Fkbp5^{+/+}$ mice display significantly increased uterine $Akr1c18$ mRNA and protein levels on E17.25, indicating that local P4 metabolism, rather than systemic P4 withdrawal, initiates parturition as decline in serum P4 levels occurs 24 h later on E18.25 (Guzeloglu-Kayisli et al, 2021). Therefore, luteolysis is likely a consequence, rather than a cause, of labor in mice, reflecting increasing uterine prostaglandin production, as supported by Roizen et al, (Roizen et al, 2008). Prior studies (Sugimoto et al, 1997) suggest that parturition is initiated when serum P4 levels fall below 10 ng/mL in mice. In the current study, we observed that restraint stressed mice had serum P4 levels below 5 ng/ml, while 15dPGJ2 administration increased these levels to over 10 ng/ml in addition to reducing $Fkbp51$ levels.

Co-administration of 15dPGJ2 + P4 further raised serum P4 levels to 36 ng/ml, reduced local uterine $Fkbp5$ levels and lowered $Ak1c18$ levels to block P4 metabolism, which collectively sustained PR activity longer than their alone administration, thereby contributing to additional extension of gestation length in restraint stressed mice. In this study, we confirmed that while serum P4 levels were lower in stressed mice compared to controls at E18.25, stress-induced PTB appears to result from premature increases of uterine $FKBP5$ levels locally in the uterus at E17.25 (Guzeloglu-Kayisli et al, 2021) and E18.25, which blocks PR action, elevates the EC50 of P4 required for PR activation and enhances Akr1c18 levels to increase P4 metabolism (Nadeem et al, 2016; Paul et al, 2023; Piekorz et al, 2005). In support, therapeutic inhibitions of $Fkbp5$ levels by 15dPGJ2 alone or in combination with P4 or R5020 or genetic ablation of $Fkbp5$ (Guzeloglu-Kayisli et al, 2021) in restrained mice hinders these changes, proving that maternal stress induced PTB is due to FKBP51-dependent early inhibition of uterine PR signaling. Besides the 15dPGJ2-meditated inhibition of $Fkbp5$, lower $Akr1c18$, and $Oxtr$ levels at E18.25 in the uteri of stressed mice also contributes to prolonged gestation in a FKBP51-dependent manner as these two labor initiating mediators are lower in restraint stressed $Fkbp5^{-/-}$ mice versus restraint stressed $Fkbp5^{+/+}$ mice (Guzeloglu-Kayisli et al, 2021).

Deficiency of $Akr1c18$ results in prolonged gestation (Ishida et al, 2007; Piekorz et al, 2005) supporting increased $Akr1c18$ contribution to molecular mechanisms of restraint stress induced PTB. Moreover, consistent with our present findings in the mouse, two groups have shown increased myometrial 20α-HSD levels in human term (Nadeem et al, 2016; Williams et al, 2012) and preterm (Nadeem et al, 2016) parturition. In our previous study, we found that stress-induced uterine Akr1c18 levels were significantly lower in stressed $Fkbp5^{-/-}$ mice vs. stressed $Fkbp5^{+/+}$ mice (Guzeloglu-Kayisli et al, 2021). In the current study, stress-induced $Akr1c18$ expression was significantly reduced in restraint-stressed mice treated with 15dPGJ2 + P4 at E18.25. This reduces uterine P4 metabolism, which lowers unliganded PR levels and blocks PR-FKBP51 binding, leading to enhanced PR-mediated transcriptional activity. These findings are consistent with those in humans, which show increased PR-FKBP51 interactions in iPTB specimens (Guzeloglu-Kayisli et al, 2021). A recent study also supports this by showing that enhanced 20α-HSD expressions reduce nuclear P4 levels, resulting in unliganded PR in the myometrium during parturition (Nadeem et al, 2016). Additionally, in the absence of P4,

FKBP51 has a higher affinity for PR, and P4 binding modulates a switch from FKBP51 to FKBP52 expression, facilitating the activation of PR (Davies et al, 2002; Nair et al, 1997; Smith et al, 1993).

Our observation that the highest levels of $Oxtr$ are in the uteri of restrained mice at E18.25, consistent with their earlier delivery, further confirms that this is a FKBP51-dependent mechanism, since uterine $Oxtr$ expression, is physiologically suppressed during pregnancy and increases near parturition in all mammalian species, including primates and ruminants (Bishop, 2013). In contrast, 15dPGJ2 alone or in combination with P4 prevented restraint-induced labor-related uterine changes, displaying lower $Oxtr$ levels. This suggests that local increases in uterine FKBP51 levels creates a pathological cycle among FKBP51, PR and Akr1c18-dependent P4 metabolism to trigger PTB and that 15dPGJ2 impedes this cycle by inhibiting induction of FKBP51 expression, which likely also enhances efficacy of exogenous P4 supplementation leading to additive effects on extending gestational length (Fig. 7).

In conclusion, our results reveal that 15dPGJ2 alone or with progestin co-treatment significantly blocks stress-induced FKBP51 levels, lowers PR-FKBP51 interactions, enhances exogenous P4 response and reduces uterine P4 metabolism, while also decreasing labor-inducing and pro-inflammatory markers, to increase pup birth weight, and most importantly, prolonged gestation as summarized in Fig. 7. These results, obtained from complementary proof-of-concept models in human decidual cell cultures and a rodent model of maternal stress-induced PTB, provide the first substantial evidence, strongly supporting the clinical use of 15dPGJ2—alone or in combination with either P4 or R5020— for treating women at risk of stress-related iPTB.

## Methods

**Reagents and tools table**

| Reagent/resource | Reference or source | Identifier or catalog number |
|---|---|---|
| **Experimental models** | | |
| Decidualized stromal cells (DSCs) | Primary decidual cells isolated from human placental samples obtained from healthy term placentas. | * |
| T47D cells | ATCC | Cat#HTB-133 |
| C57BL/6 mice (*M. musculus*) | In house inbreed wild-type mice | * |
| **Antibodies** | | |
| Goat anti-FKBP51 | R&D systems | Cat #AF4094 |
| Rabbit anti-PR-A/B | Cell Signaling | Cat #8757S |
| Rabbit IgG isotype | Cell signaling | Cat #3900S |
| Goat IgG isotype | Vector Lab. | Cat #I-5000 |
| Goat anti rabbit-peroxidase (secondary antibody) | Vector Lab. | Cat #PI-1000 |
| Horse anti goat-peroxidase (secondary antibody) | Vector Lab. | Cat #PI-9500 |

| Reagent/resource | Reference or source | Identifier or catalog number |
|---|---|---|
| **Oligonucleotides and other sequence-based reagents** | | |
| Human TaqMan FAM labeled *FKBP5* Gene Expression Assay | Thermo Fisher Scientific | Hs01561006_m1 Cat# 4331182 |
| Human TaqMan FAM labeled *IL1B* Gene Expression Assay | Thermo Fisher Scientific | Hs01555410_m1 Cat #4453320 |
| Human TaqMan FAM labeled *PTGS2* Gene Expression Assay | Thermo Fisher Scientific | Hs00153133_m1 Cat #4453320 |
| Human TaqMan FAM labeled *OXTR* Gene Expression Assay | Thermo Fisher Scientific | Hs00168573_m1 Cat # 4331182 |
| Human TaqMan FAM labeled *ACTB* Gene Expression Assay | Thermo Fisher Scientific | Hs99999903-m1 Cat # 4331182 |
| Mouse TaqMan FAM labeled *Fkbp5* Gene Expression Assay | Thermo Fisher Scientific | Mm00487406_m1 Cat # 4453320 |
| Mouse TaqMan FAM labeled *Oxtr* Gene Expression Assay | Thermo Fisher Scientific | Mm01182684-m1 Cat # 4448892 |
| Mouse TaqMan FAM labeled *Akr1c18* Gene Expression Assay | Thermo Fisher Scientific | Mm00506289_m1 Cat # 4448892 |
| Mouse TaqMan FAM labeled *Il1b* Gene Expression Assay | Thermo Fisher Scientific | Mm00434228_m1 Cat # 4453320 |
| Mouse TaqMan FAM labeled *Ptgs2* Gene Expression Assay | Thermo Fisher Scientific | Mm00478374_m1 Cat # 4453320 |
| MouseTaqMan FAM labeled *Actb* Gene Expression Assay | Thermo Fisher Scientific | Mm00607939-s1 Cat # 4331182 |
| **Chemicals, enzymes and other reagents** | | |
| Dulbecco's MEM/F12 Media | Thermo Fisher Scientific | Cat # 11039-021 |
| Bovine Calf Serum | GeminiBio | Cat # 100-506-500 |
| Antibiotic/Antimycotic (100x) | Thermo Fisher Scientific | Cat #15240062 |
| Estradiol | Sigma-Aldrich | Cat #E2758 |
| Medroxyprogesterone acetate | Sigma-Aldrich | Cat #M1629 |
| Forskolin | Tocris | Cat # 1099 |
| 15dPGJ2 | Millipore-Sigma | Cat #538927 |
| 9 Nitrooleate acid | Cayman | Cat #10008042 |
| 10 Nitrooleate acid | Cayman | Cat #10008043 |
| Progesterone (P4) | Sigma-Aldrich | Cat #P8783 |
| Promegestone (R5020) | Perkin Elmer | Cat #NLP004005 |
| Dexamethasone (Dex) | Sigma-Aldrich | Cat #D4902 |
| Nuclear Complex Co-IP kit | Active Motif | Cat #54001 |
| DC protein quantification assay | Bio-Rad | Cat #5000116 |

| Reagent/resource | Reference or source | Identifier or catalog number |
|---|---|---|
| Protease/Phosphatase inhibitor cocktail (100X) | Cell Signaling | Cat#5872 |
| Magnetic Protein A/G beads | Thermo Fisher Scientific | Cat #88803 |
| 2x Laemmli Sample Buffer | Bio-Rad | Cat #1610737 |
| 10% Tris-HCl gel | Bio-Rad | Cat #4561034 |
| 7.5% Tris-HCl gel | Bio-Rad | Cat #4561024 |
| Non-fat dry milk | Cell Signaling | Cat #9999S |
| Tween-20 | Thermo Fisher Scientific | Cat # J20005AP |
| Nitrocellulose membrane | Bio-Rad | Cat #1620112 |
| ECL substrate | Thermo Fisher Scientific | Cat #32106 |
| RNeasy Mini kit | Qiagen | Cat #74104 |
| RNase-Free DNase Set | Qiagen | Cat #79256 |
| Omniscript RT Kit | Qiagen | Ca #205113 |
| PGF$_{2\alpha}$ ELISA Kit | Cayman Chem | Cat # 516011 |
| Corticosterone ELISA Kit | R&D | Cat # KGE009 |
| P4 ELISA Kit | BioVendor | Cat #RTC008R |
| MycoStrip Mycoplasma Detection Kit | Invivogen | Cat#rep-mys-20 |
| **Software** | | |
| ZEN Imaging system | Zeiss | |
| Image J 1.52a | National Institutes of Health, Bethesda, MD | |
| SigmaPlot v11.0 | Systat Software Inc. | |
| **Other** | | |
| Light Microscope | Zeiss | Axio-Imager A2 |
| Microplate reader | Bio-Rad iMARK | Serial no#19638 |
| ChemiDoc MP Imaging system | Bio-Rad | Serial no#734BR4295 |
| Standard restraint chambers (1.5" × 4") | Braintree Scientific Inc. | Cat#FB-P S-PL |

## Study approval

Collection of human placental specimens for isolation of decidual stromal cells was approved by the University of South Florida Institutional Review Board (#19472). Written informed consent was obtained from patients before inclusion in the study and the experiments conformed to the principles set out in the WMA Declaration of Helsinki and the Department of Health and Human Services Belmont Report. All patients are female, with an average age of $28.16 \pm 3.54$ years (mean ± s.d.). The patient group was equally divided by ethnicity, with 50% identifying as Hispanic and 50% as non-Hispanic. All other patient information was de-identified. Animal breeding and experimental procedures were conducted following approval from the Animal Care and Use Committee at the University of South Florida (USF#10574 R and #10898).

## Cell culture

Term DSCs ($n = 6$) were isolated as described (Guzeloglu-Kayisli et al, 2012; Lockwood et al, 2012) from uncomplicated pregnancies undergoing repeat cesarean delivery at term. Term DSCs were cultured in basal medium containing a phenol red-free 1:1 v/v of Dulbecco's MEM/F12 (Thermo Fisher Scientific, Waltham, MA) supplemented with 5% charcoal stripped bovine calf serum (Gemini-Bio, West Sacramento, CA) and 1% antibiotic/antimycotic (Thermo-Fisher). At 60% confluency, cells were in vitro decidualized for 7 days in basal media with $10^{-8}$ M E$_2$ (Sigma-Aldrich) + $10^{-7}$ M medroxyprogesterone acetate (MPA; Sigma-Aldrich) + $5 \times 10^{-7}$ M forskolin (Tocris, Minneapolis, MN). Then, cultures were switched to a serum-free defined media as described (Lockwood et al, 2010) for 24 h, then incubated with the following treatments: (1) $10^{-8}$ M E$_2$ alone; (2) E$_2$ + $10^{-5}$ M 15dPGJ2 (Millipore); (3) E$_2$ + $10^{-7}$ M P4 (Sigma-Aldrich); (4) E$_2$ + $10^{-7}$ M R5020 (Perkin Elmer); (5) E$_2$ + P4 + 15 dPGJ2; and (6) E$_2$ + R5020 + 15dPGJ2 for 6 and 24 h. In parallel, a second set of cultures were pre-treated 24 h with E$_2$ + $10^{-7}$ M Dex (Sigma-Aldrich) to induce endogenous FKBP51 levels, then incubated with same steroid treatments as mentioned above. CMS from 24 h cultures were collected and stored at $-80\,°C$ for ELISA and Mass-Spectrometry analysis. In the final set of experiments, term DSCs were treated with 0.1, 1, 2.5, 5 and 10 μM 15dPGJ2 for 6 h to assess its dose dependent effects. T47D cells ($n = 4$) were cultured in basal media until reaching 70% confluency. Then, cells were treated with $10^{-8}$ M E$_2$ ± $10^{-7}$ M P4 or $10^{-7}$ M R5020 for 24 h, followed by $10^{-5}$ M 15dPGJ2 ± E$_2$ or P4 or R5020 treatments for an additional 24 h. Cells were then washed with cold PBS and immediately processed for cytoplasmic-nuclear protein extraction, as described below.

## Cytoplasmic and nuclear cell extraction

Cytoplasmic and nuclear extracts were prepared using Nuclear Complex Co-IP kit (Active Motif, Carlsbad, CA) per manufacturer's instructions for immunoprecipitation assay as described (DeCaprio and Kohl, 2017). Briefly, after trypsinization and centrifugation, pellets were lysed with 500 μl hypotonic buffer supplemented with protease/phosphatase inhibitor cocktail (Cell Signaling), and 0.5 mM PMSF for 15 min on ice followed by centrifugation at $6200 \times g$ for 1 min at 4 °C. Supernatant was collected as the cytoplasmic extract. The cell pellets were washed and incubated in 200 μl of complete digestion buffer containing phosphatase/protease inhibitor cocktail, PMSF and enzymatic shearing cocktail to release undissociated protein complexes from the DNA for 1 h on ice. Nuclear extracts were obtained after centrifugation at $18,000 \times g$ for 10 min at 4 °C. The concentration of cytoplasmic and nuclear extract was quantified by DC protein quantification assay (Bio-Rad, Hercules, CA) at 750 nm with a microplate reader (Bio-Rad) and stored at $-80\,°C$ for immunoprecipitation analysis.

## Immunoprecipitation

In all, 200 μg cytoplasmic or nuclear extracts were precleared with 25 μl pre-wash magnetic Protein A/G beads (10 mg/ml; Thermo-Fisher) to reduce non-specific protein binding for 1 h at 4 °C. Samples were then incubated with either goat anti-FKBP51 (1.5 μg, R&D) or rabbit anti-PR-A/B (0.7 μg Cell Signaling) antibody or host specific non-immune IgG isotype (Cell Signaling) as a negative

control for 24 h at +4 °C to form immunocomplex on shaker. Next day, samples were incubated with prewashed 25 μl of magnetic Protein A/G beads for 2 h at room temperature with gentle agitation. After bead pellet discarded, supernatant of immunoprecipitated complexes was eluted with 40-μl 2x reducing loading buffer (Bio-Rad) and processed for immunoblotting.

## Immunoblotting

Immunoprecipitated complexes were subjected to reducing SDS-PAGE on a 7% or 10% Tris-HCl gel (Bio-Rad) and transferred onto nitrocellulose membranes (Bio-Rad) as described (Guzeloglu-Kayisli et al, 2015). Following blocking with 5% non-fat milk in Tris-buffered saline with 0.1% Tween-20 (TBS-T) for 1 h at RT, the membrane was incubated with goat polyclonal FKBP51 (1/700, R&D) or rabbit PR-A/B (1/500, Cell Signaling) primary antibodies, overnight at 4 °C. After several washes with TBS-T, the membranes were incubated with appropriate secondary antibodies (1/5000, Vector Lab, Newark, CA) for 1 h at RT. Following washing steps, the immunoblots were developed with ECL Substrate (Thermo-Fisher) visualized using chemiluminescent imager (Bio-Rad).

## RNA isolation and quantitative real-time PCR (qPCR)

Total RNAs from cultured term DSCs and uterine tissues collected at E18.25 were isolated using RNeasy Mini kit (Qiagen, Valencia, CA) as described (Guzeloglu-Kayisli et al, 2015; Guzeloglu-Kayisli et al, 2021). After DNase treatment, 1 μg of total RNA was reverse transcribed with random decamers at 37 °C for 1 h using the Omniscript RT Kit (Qiagen). Analysis of qPCR was used to quantify expression levels of: (1) *FKBP5; IL1B, PTGS2 and OXTR*; and (2) *Fkbp5, Oxtr, Akr1c18, Il1b,* and *Ptgs2* by using human or mouse gene specific FAM-labelled TaqMan gene expression assays (Thermo-Fisher) on ABI 7500 thermocycler instrument. The $2^{-\Delta\Delta Ct}$ (cycle threshold) method was used to calculate relative expression levels after normalization to *ACTB or Actb* mRNA levels as an internal control.

## Ultraperformance liquid chromatography-tandem mass spectrometry (UPLC/MS)

CMS collected from 24-h term DSC cultures ($n = 6$) were analyzed for metabolomic profiling to assess the levels of P4 and its metabolites by Creative Proteomics (Shirley, NY) using UPLC/MS. A mixed standard solution of all targeted steroid compounds was prepared in an internal standard (IS) solution equipped with methanol for each isotope-labeled steroid hormone. This standard solution was serially diluted in the same IS solution to create 9 calibration solutions. For analysis, 200 μL CMS sample mixed with 100 μL of IS solution was combined with 1.7 mL of water in 3-mL polymeric reversed-phase SPE cartridges (200 mg). After mixing thoroughly, the flow-through fractions were discarded under positive low pressure. The cartridges were washed with 3 mL of water, and the steroids were eluted with 2 mL methanol into 3-mL glass test tubes. The methanol was evaporated using a speed-vacuum concentrator, and the residues were dissolved in 100 μL methanol. Twenty microliters of each solution were injected into a C18 column for UPLC/MS analysis on an Agilent 1290 UHPLC system coupled with a Sciex 7500 QQQ mass spectrometer,

operating in positive-ion detection mode. The mobile phase consisted of 0.05% formic acid in water and 0.05% formic acid in a methanol-acetonitrile mixture, used for binary solvent gradient elution under optimized separation and detection conditions. Linear-regression calibration curves were constructed using data from the calibration solutions. The concentrations of the detected compounds were determined by interpolating these calibration curves with the analyte-to-IS peak area ratios measured from the sample solutions.

## ELISA

$PGF_{2\alpha}$ levels were measured in collected CMS using ELISA kit per manufacturer's protocol (Cayman Chem) and read at 415 nm using a microplate reader (Bio-Rad) as described (Guzeloglu-Kayisli et al, 2015). In mice, serum corticosterone (BioVendor, Ashville, NC) and P4 levels (R&D) were measured using EIA kits as we described (Guzeloglu-Kayisli et al, 2021).

## Mouse model for stress-induced PTB

All mice were housed in a temperature-controlled environment with 12-h light/dark cycles (6 am/6 pm) with ad libitum access to food and water. In all, 6–8-week C57BL/6 adult female mice were mated with 12–14-week-old males by housing 1:1 for 4 h (9.00 am–1.00 pm). The presence of vaginal plug confirmed pregnancy, and the gestational day was designated as 0 dpc at that time. Starting on E8 through E18, pregnant mice were daily exposed to restraint stress in three sessions each for 1 h (total 3 h per day) using standard restraint chambers (1.5" × 4"; Braintree Scientific Inc., Braintree, MA). The stress schedule was varied day to day to reduce possible restraint stress habituation.

## 15dPGJ2 and/or progestins administration

15dPGJ2 (Millipore), P4 (Sigma-Aldrich) or R5020 (Perkin Elmer) was dissolved in ethanol and stored at −20 °C. 15dPGJ2 was diluted daily at a concentration of 20 µg in 100 µl in PBS, while P4 or R5020 was diluted at concentration 0.2 mg in 100 µl in coconut oil. Starting at E14 through E18, restraint pregnant mice were randomly assigned into six groups to receive injections of: (1) i.p. and s.c. placebo; (2) i.p. 15dPGJ2 (20 µg/dam); (3) s.c. P4 (0.2 mg/dam); (4) i.p.15dPGJ2 (20 µg/dam) + s.c. P4 (0.2 mg/dam); (5) s.c. R5020 (0.2 mg/dam) injection; and (6) i.p. 15dPGJ2 (20 µg/dam) + s.c. R5020 (0.2 mg/dam). Each injection was administered at 9 am, prior to the first stress exposure. Undisturbed pregnant mice were used as control to determine natural gestational length and to compare with restraint stress/placebo and other treated groups.

## Monitoring labor time and litter size

The delivery timing was detected by monitoring mice from E18 through E22 every 4 h until completion of delivery and gestational length was calculated. We previously reported a gestational length of 19.2 ± 0.1 days (mean ± SEM) in these mice under undisturbed conditions (Guzeloglu-Kayisli et al, 2021). Thus, in any experiential groups, delivery earlier than E19.0 is defined as PTB, between ≥E19.0 to ≤E19.75 accepted as term birth; and after E19.75 accepted as delayed parturition. The litter size with live and dead pups after completion of labor were recorded to evaluate any side effect resulting from any treatments.

## Monitoring birth and postnatal weight

To determine the effects of maternal restraint stress and other treatments on birth weight, pup weight was recorded at birth immediately after completion of labor and postnatal day 21 before weaning.

## Blood sample collection

Blood samples (~100 µl) were obtained by cardiac puncture under general anesthesia on E18.25. Serum samples separated by centrifugation were stored at −80 °C.

## Histopathological evaluation

Pups ($n = 5$/sex) were euthanized on postnatal day 25. The uterus, ovary, testis, and vital organs, including the brain, kidney, and heart were harvested and fixed in %4 paraformaldehyde. After the tissues were embedded in paraffin and sectioned, they were stained with H&E for histopathological evaluation. The evaluation was conducted by researchers blinded to experimental groups to prevent bias.

## Fertility assessment

No effects of P4 use during pregnancy on offspring fertility have been reported (https://www.drugs.com/pregnancy/progesterone.html). Since the potential side effects of 15 dPGJ2 on fertility are unknown, we conducted a continuous mating study over 6 weeks, as described previously (Guzeloglu-Kayisli et al, 2012) on offspring exposed to prenatal maternal stress from E8 to E18 and/ or received i.p. 15 dPGJ2 injection from E14 to E18.

## Statistical analysis

SigmaPlot version 11.0 software (Systat Software, San Jose, CA) was used for statistical analysis. Kolmogorov–Smirnov test assessed data for normality. Pair-wise multiple comparisons by one-way ANOVA were followed by the post-hoc Tukey test or Student–Newman–Keuls test or Dunn's method for analysis of data with parametric or non-parametric distribution, respectively. In two groups comparison, a $t$ test or a Mann–Whitney $U$ test was used for analysis of data with parametric or non-parametric distribution, respectively.

Sample size calculation in in vitro studies based on our qPCR assessment of *FKBP5* expression in DSC cultures, in which we observed that *FKBP5* mRNA levels decrease by ~50% by 15PGJ2+Dex vs. Dex alone treatment (s.d.: ~20%; $P < 0.01$). Thus, to achieve 90% power with an alpha (type I error) of 0.01, and assuming the same standard deviation, 5 biological repeats of DSC cultures are required to achieve significance. Sample size calculation in mice studies is based on our previous study (Guzeloglu-Kayisli et al, 2021) in which wild-type mice are reported to have a gestational length of 19.2 ± 0.6 days (mean ± s.d.) and 18.7 ± 0.1 days under restraint stress. In preliminary results, 15 dPGJ2 administration to stressed mice significantly improved

## The paper explained

### Problem

Preterm birth (PTB), defined as delivery before 37 weeks of gestation, affects 10.5% of all U.S. births, with a higher rate of 14–15% among Black women. The stress response co-chaperone FKBP51 binds to progesterone receptors (PRs), inhibiting PR-mediated transcriptional activity, which leads to functional progesterone withdrawal and results in stress-associated and idiopathic PTB. This study investigates the efficacy of 15dPGJ2, alone or in combination with progesterone or synthetic progestin R5020, in preventing maternal stress-induced PTB and evaluates its potential as anti-PTB therapeutic.

### Results

We found that 15dPGJ2 prevented maternal stress-induced PTB in mice by reducing levels of *Fkbp5* and labor-inducing mediators. 15dPGJ2 co-treatment with progestin, further extended gestation, suggesting 15dPGJ2 as a promising anti-PTB therapy.

### Impact

Our findings demonstrate that 15dPGJ2 targets FKBP51, a key regulator of the stress response pathway, significantly inhibiting its levels in a safe and effective manner. This therapy has the potential to fundamentally change clinical practice and reduce PTBs. Inhibition of uterine FKBP51 activity could also have broad applicability in other causes of PTB. Furthermore, elevated FKBP51 levels are linked to stress-related psychiatric disorders, type 2 diabetes, and obesity, suggesting that targeting FKBP51 could offer therapeutic benefits beyond obstetrics.

gestational length (mean ± s.d. $19.27 \pm 0.52$ days, $P < 0.05$) and permitted 62.5% of restrained mice to deliver on day 19.0 or thereafter. Thus, to achieve 80% power with an alpha (type I error) of 0.05, and assuming the same s.d. detected in 15dPGJ2 administration in restrained mice, 11 mice per group are required to achieve significance.

## Data availability

This study includes no data deposited in external repositories

The source data of this paper are collected in the following database record: biostudies:S-SCDT-10_1038-S44321-025-00211-9.

## Peer review information

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

## Acknowledgements

This study was supported by NIH/NICHD grant# R41 HD113465-01 to David friend and Ozlem Guzeloglu-Kayisli. The authors thank Kristin Scott and Emily McMains (Grant Engine) for their contributions to the overall discussions and the design of Fig. 7, and Kellie Larsen for her assistance with stress application.

## Author contributions

**Ozlem Guzeloglu-Kayisli**: Conceptualization; Data curation; Formal analysis; Supervision; Funding acquisition; Validation; Investigation; Visualization; Methodology; Writing—original draft; Writing—review and editing. **Asli Ozmen**: Resources; Data curation; Formal analysis; Supervision; Validation; Investigation; Visualization; Writing—original draft; Writing—review and editing. **Busra Cetinkaya Un**: Formal analysis; Supervision; Validation; Investigation; Visualization; Writing—review and editing. **Burak Un**: Formal analysis; Validation; Investigation; Visualization; Writing—review and editing. **Jacqueline Blas**: Investigation; Writing—review and editing. **Isabella Johnson**: Resources; Methodology; Writing—review and editing. **Andrea Thurman**: Methodology; Writing—review and editing. **Mark Walters**: Methodology; Writing—review and editing. **David Friend**: Funding acquisition; Methodology; Writing—review and editing. **Umit A Kayisli**: Conceptualization; Data curation; Formal analysis; Supervision; Funding acquisition; Validation; Visualization; Methodology; Writing—original draft; Writing—review and editing. **Charles J Lockwood**: Conceptualization; Data curation; Formal analysis; Funding acquisition; Validation; Visualization; Methodology; Writing—original draft; Writing—review and editing.

Source data underlying figure panels in this paper may have individual authorship assigned. Where available, figure panel/source data authorship is listed in the following database record: biostudies:S-SCDT-10_1038-S44321-025-00211-9.

## Disclosure and competing interests statement

Ozlem Guzeloglu-Kayisli, Umit A Kayisli, and Charles J Lockwood are inventors on patent application that includes use of 15dPGJ2 in preventing preterm birth. sabella Johnson is the Senior Manager of Product Development at Dare Bioscience, Andrea Thurman is a Medical Director of Dare Bioscience, Mark Walters is a co-founder and serves as the Vice President of Operations at Dare Bioscience, and David Friend is the Chief Scientific Officer of Dare Bioscience. The authors declare no competing interests.

# Expanded View Figures

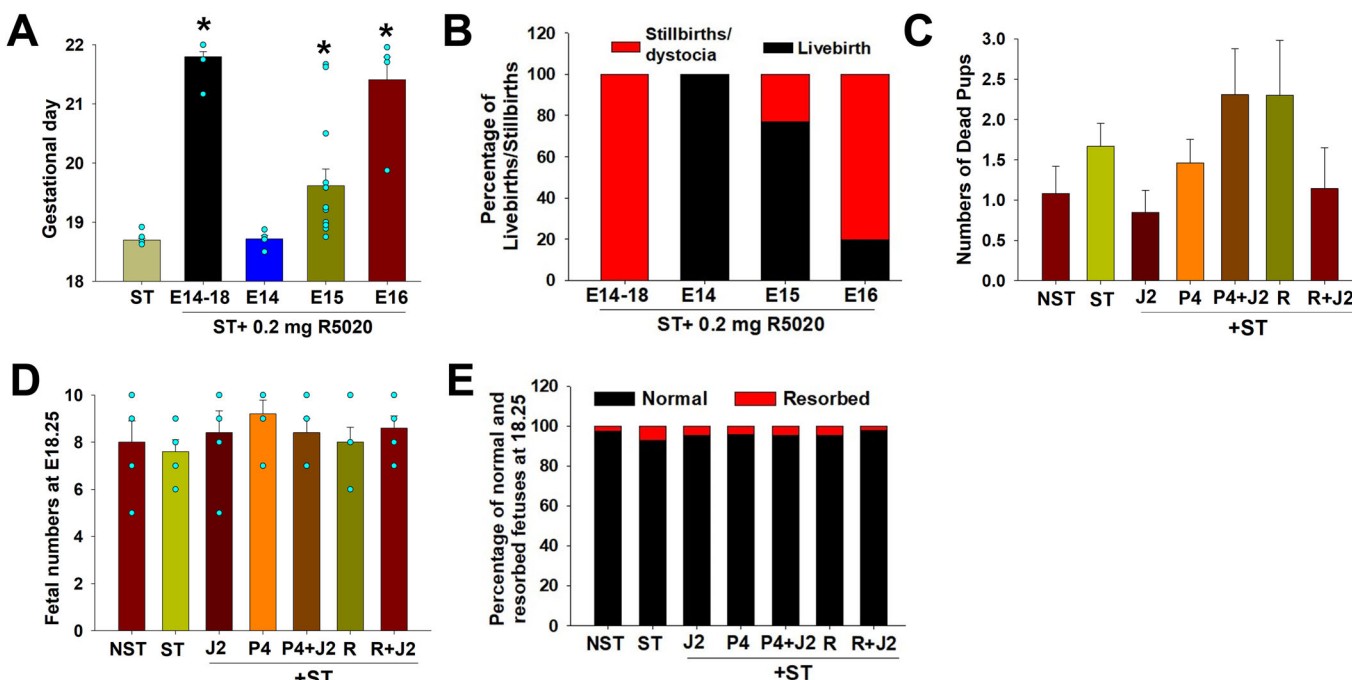

**Figure EV1. Gestation day dependent effects of R5020 injections in preventing maternal stress induced PTB.**

(A, B) Gestational length (A) and percentage of live and still births (B) in time-pregnant wild-type mice subjected to maternal restraint stress (ST, $n = 12$) between E8 and 18 and received s.c. injection of 0.2 mg/dam R5020 from E14 to 18 ($n = 9$), or only on E14 ($n = 5$), only on E15 ($n = 13$) or only on E16 ($n = 5$). Bars represent mean ± s.e.m. (A) *$P < 0.05$ vs. ST, analyzed with one-way ANOVA followed by Dunn's method. (C) Number of dead pups in mice left undisturbed state as control (NST, $n = 12$) or subjected to restraint stress between E8 and E18 and received from E14 to 18 injection of placebo (ST, $n = 12$) ± 15dPGJ2 (J2, $n = 13$), or P4 (P4, $n = 13$) or P4 + 15dPGJ2 (P4 + J2, $n = 13$), or R5020 (R, $n = 10$) or R5020 + 15dPGJ2 (R + J2, $n = 8$). Bars represent mean ± s.e.m. $P = 0.09$, analyzed by One-Way ANOVA. (D, E) Total fetus numbers (D) and percentage of normal and resorbed fetuses (E) at E18.25 in NST or ST or J2 or P4 or P4 + J2 or R or R + J2. Bars represent mean ± s.e.m.; $n = 5$ biological replicates. (D) $P = 0.7$, analyzed by One-Way ANOVA. Note that R5020 was only single injection given on E15 (C, D).

                                                                 

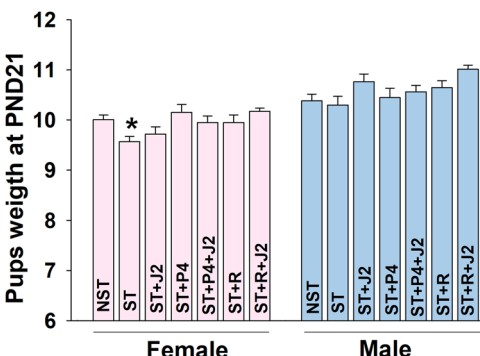

**Figure EV2.  Impact of prenatal stress and 15dPGJ2, alone or in combination with P4 or R5020 on the weight of female and male pups at postnatal day 21 (PND21).**

Body weight were measured on PND21 for both female and male offspring mice from control (NST) or subjected to maternal restraint stress between E8-18 and received from E14 to 18 injection of placebo (ST), or +15dPGJ2 (ST + J2), or +P4 (ST + P4) or +P4 + 15dPGJ2 (ST + P4 + J2), or +R5020 (ST + R) or +R5020 + 15dPGJ2 (ST + R + PJ2). Bars represent mean ± s.e.m.; *P = 0.005 vs. NST, analyzed by Mann–Whitney $U$ test. Total number of female pups measured: NST $n = 44$, ST $n = 32$; J2 $n = 38$; P4 $n = 37$; J2 + P4 $n = 30$; R $n = 20$; and J2 + R $n = 23$; male pups measured: NST $n = 26$, ST $n = 27$; J2 $n = 43$; P4 $n = 33$; J2 + P4 $n = 32$; R $n = 27$; and J2 + R $n = 28$. Note that R5020 was only single injection given on E15.

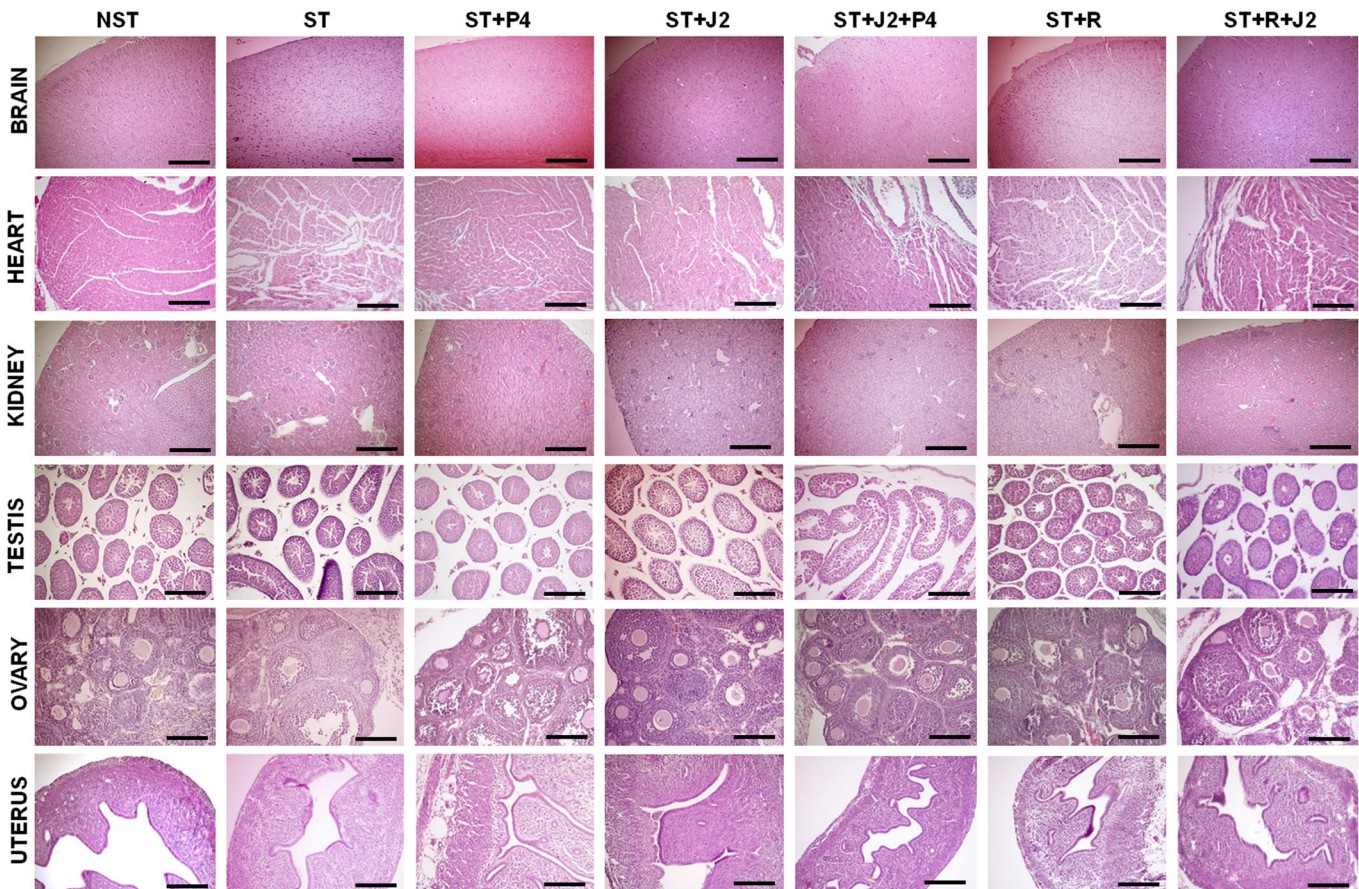

**Figure EV3.   Histologic comparison of the brain, heart, kidney, testis, ovary, and uterus on postnatal day 21 (PND21).**

H + E stained tissues were compared for any signs of hemorrhage, fibrotic or necrotic areas and/or inflammatory leukocyte infiltration. Representative pictures of H + E stained tissues obtained from pups of time-pregnant wild-type mice left undisturbed state as control (NST) or subjected to maternal restraint stress and received injection of placebo (ST), or +15dPGJ2 (ST + J2), or +P4 (ST + P4) or +P4 + 15dPGJ2 (ST + P4 + J2), or +R5020 (ST + R) or +R5020 + 15dPGJ2 (ST + R + PJ2). *N* = 5 biological replicates. Scale bars = 300 μm.

