## [Peer Review File · EMBO Molecular Medicine]

Targeting FKBP51 prevents stress-induced preterm birth

Ozlem Guzeloglu Kayisli, Asli Ozmen, Busra Cetinkaya-Un, Burak Un, Jacqueline Blas, Isabella Johnson, Andrea Thurman, Mark Walters, David Friend, Umit Kayisli, and Charles Lockwood

Corresponding author(s): Ozlem Guzeloglu Kayisli (ozlem2@usf.edu) , Charles Lockwood (cjlockwood@usf.edu)

Review Timeline:

Submission Date:	21st Dec 24
Editorial Decision:	21st Jan 25
Revision Received:	7th Feb 25
Editorial Decision:	19th Feb 25
Revision Received:	23rd Feb 25
Accepted:	24th Feb 25

Editor: Zeljko Durdevic

Transaction Report:

21st Jan 2025

Dear Dr. Guzeloglu Kayisli,

Thank you for the submission of your manuscript to EMBO Molecular Medicine. We have now received feedback from the two reviewers who agreed to evaluate your manuscript. Both referees recognize interest of the study but also raise important concerns that should be addressed in a major revision. If you would like to discuss further the points raised by the referees, I am available to do so via email or video. Let me know if you are interested in this option.

We would welcome the submission of a revised version within three months for further consideration. Please let us know if you require longer to complete the revision.

I look forward to seeing a revised form of your manuscript as soon as possible.

I look forward to receiving your revised manuscript.

Yours sincerely,

Zeljko Durdevic

We require:

- 1) A .docx formatted version of the manuscript text (including legends for main figures, EV figures and tables). Please make sure that the changes are highlighted to be clearly visible.
- 2) Individual production quality figure files as .eps, .tif, .jpg (one file per figure). For guidance, download the 'Figure Guide PDF': (<https://www.embopress.org/page/journal/17574684/authorguide#figureformat>).
- 3) A .docx formatted letter INCLUDING the reviewers' reports and your detailed point-by-point responses to their comments. As part of the EMBO Press transparent editorial process, the point-by-point response is part of the Review Process File (RPF), which will be published alongside your paper.
- 4) A complete author checklist, which you can download from our author guidelines (<https://www.embopress.org/page/journal/17574684/authorguide#submissionofrevisions>). Please insert information in the checklist that is also reflected in the manuscript. The completed author checklist will also be part of the RPF.
- 5) Please note that all corresponding authors are required to supply an ORCID ID for their name upon submission of a revised manuscript.
- 6) It is mandatory to include a 'Data Availability' section after the Materials and Methods. Before submitting your revision, primary datasets produced in this study need to be deposited in an appropriate public database, and the accession numbers and

database listed under 'Data Availability'. Please remember to provide a reviewer password if the datasets are not yet public (see <https://www.embopress.org/page/journal/17574684/authorguide#dataavailability>).

12) Author contributions: You will be asked to provide CRediT (Contributor Role Taxonomy) terms in the submission system. These replace a narrative author contribution section in the manuscript.

13) A Conflict of Interest statement should be provided in the main text.

14) Every published paper now includes a 'Synopsis' to further enhance discoverability. Synopses are displayed on the journal webpage and are freely accessible to all readers. They include a short stand first (maximum of 300 characters, including space) as well as 2-5 one-sentences bullet points that summarizes the paper. Please write the bullet points to summarize the key NEW findings. They should be designed to be complementary to the abstract - i.e. not repeat the same text. We encourage inclusion of key acronyms and quantitative information (maximum of 30 words / bullet point). Please use the passive voice. Please attach

these in a separate file or send them by email, we will incorporate them accordingly.

15) Include a Reagents and Tools Table as part of the Methods section, which can be downloaded from our author guidelines (<https://www.embopress.org/page/journal/17574684/authorguide#structuredmethods>)

***** Reviewer's comments *****

Referee #1 (Comments on Novelty/Model System for Author):

The technical quality is acceptable, although the quality of the western blots is not very high.

There are many reports focused on drug therapy that prevents preterm birth in mouse models. The model presented here is novel, but one wonders if this model is any more translatable than any of the others.

Referee #1 (Remarks for Author):

This is an interesting manuscript that addresses a pressing clinical issue. I have a few major questions and then some minor concerns.

Major Issues:

The experiment in depicted in Figure 4 shows that J2 does not affect the metabolism of P4. Moreover, in Figure 7C, neither J2 nor P4 affects levels of Akr1C18, an enzyme involved in P4 metabolism. Why then does the combination of J2 and P4 cause a significant drop in Akr1C18?

Why would restraint stress not significantly increase corticosterone levels at E18.5, while it does at earlier points in gestation? Please clarify the hypothesis that restraint stress accelerates the loss of the corpus luteum without increasing corticosterone levels.

There are many publications now showing how various pharmacotherapeutic approaches delay or prevent preterm birth in mice, e.g., agents that block IL-1, compounds that inhibit NF- κ B (our work) or antioxidants to reduce oxidative stress. Yet, as you mention in this manuscript, there is currently no FDA approved drug therapy to prevent preterm birth. What is different about the approach described in this manuscript that makes it more translatable and more likely to have a clinical impact than other drug therapies that have been explored?

Minor Issues:

The figures appear stretched. Please provide better quality figures.

Indicate statistical significance in Figure 1B.

Figure 2 does not indicate which panels are B and C.

DSC stands for decidualized stromal cells. Therefore, is it redundant to say "decidualized DSC," (bottom of page 5)? Also, once the abbreviation is established, "decidualized stromal cells" should not be written out again (middle of page 7).

There is a critical typo in the first full paragraph on Page 15! (I believe 15dPG2 should be 15dPJ2.)

Centrifugation speeds should be expressed as how many g, not rpm.

There is no reference to Figure 8 in the text. Also, Figure 8 is out of focus in the copy provided.

Referee #2 (Comments on Novelty/Model System for Author):

Need clarity on why term decidual cultures were used to evaluate the impact on preterm birth

Referee #2 (Remarks for Author):

Guzeloglu-Kayisli and colleagues evaluate the role of 15dPGJ2 in mediating stress-induced preterm birth via the FKBP51-progesterone receptor.

Guzeloglu-Kayisli and colleagues build on their prior work and have performed extensive studies on FKBP51 and the use of 15dPGJ2 to reduce preterm birth. Their work is elegant and culminates in a conceptual figure for a novel method to impact preterm birth. They demonstrate that 15dPGJ2/P4 co treatment reduces both FKBP51 and AKR1C18 levels, repressing labor-inducing genes.

The manuscript may benefit from addressing these items:

The underlying biologic premise is linking maternal stress-related disorders to iatrogenic preterm birth and exploring the role of stress-mediated proteins and receptors in the pathway for preterm birth prevention. Maternal stress related disorders are listed as depression and fetal stress-related conditions including abnormal placentation. This is a broad categorization. Iatrogenic preterm birth likely has multiple etiologies including infectious, inflammatory, stress-related, genetic, etc. It is not clear that direct correlation of these findings truly impacts this broad category of preterm birth.

The withdrawal of Makena is highlighted as a major rationale for this investigation. This work builds on prior work by this group exploring the role of FKBP5 in preterm birth. The withdrawal of Makena is a distractor. Even were it still approved, other novel approaches to impact preterm birth are needed.

The evaluations of 15dPGJ2 - with fatty acid derivatives inhibit glucocorticoid induced FKBP51 levels, on levels of labor initiating molecules, and others all use term DSC cultures - why were they not evaluated in preterm DSC cultures? As pregnancy advances there are remarkable changes in the maternal, fetal, placental, and uterine physiology and metabolism. The decidua changes and has a different composition of immune and inflammatory markers. Why not perform these studies in preterm decidua?

The mouse model utilized restraint stress between E8 and E18 in three sessions for one hour - does this mean three sessions each day? Or three sessions of one hour at some timepoint between E8 and E18? Did they have access to food and water while restrained?

How were the sample sizes determined? I do not see that in the statistical analysis section.

Point to point response to Reviewers:

We sincerely appreciate the reviewers' thoughtful and constructive comments, which have enhanced the quality and clarity of our manuscript. Below, we present each reviewer's comment in bold, with our detailed responses in red, addressing them point by point.

Responses to Comments by Referee #1 (Comments on Novelty/Model System for Author):

The technical quality is acceptable, although the quality of the western blots is not very high.

Response: In the revised version of the manuscript, the quality of the immunoblot has been improved by providing higher resolution of images. Also, whole experimental repeats of immunoblots are provided as a Source data.

There are many reports focused on drug therapy that prevents preterm birth in mouse models. The model presented here is novel, but one wonders if this model is any more translatable than any of the others.

Response: Drug development in obstetrics, and specifically for preterm birth (PTB), is virtually non-existent. Several factors contribute to this stagnation, including regulatory hurdles, medico-legal hesitance regarding drug development for pregnant patients, high development costs, and the narrow market opportunity limited by the patent life of products. These statements have been briefly included in the discussion section of the revised manuscript (**Page 13**).

We patented use of 15dPGJ2 (Patent# US2022/0088032 (2022) "*Compositions and Uses Thereof for Treatment of Idiopathic Preterm Birth*") as a therapy for preventing of idiopathic PTB and have an agreement with Dare Bioscience Inc. for potential clinical translation. The findings presented in the manuscript are part of an ongoing research effort supported by an NIH STTR Phase I grant. Our successful Phase I results have laid a solid foundation for ongoing development of 15dPGJ2 + progesterone (P4), which we are formulating as a vaginal insert. In our proposed Phase II studies, we will focus on refining the duration and dosage of 15dPGJ2+P4 and defining its pharmacokinetic, safety, and toxicity profile in both pregnant and non-pregnant animal models for pre-clinical trials. After completing Phase II milestones, we plan to present our findings to the FDA during a pre-Investigational New Drug (IND) meeting for guidance on remaining pre-IND studies. We will be pursuing an initial indication for women with a singleton pregnancy and previous PTB or with a short cervix prior to 24 weeks without evidence of infection or abruption, groups at high risk for spontaneous idiopathic PTB (Frey, McLaughlin et al., 2022). Therefore, this approach builds on existing research and could provide a promising alternative to current strategies for preventing PTB. By preventing stress-induced PTB and extending gestation, this product has the potential to significantly improve neonatal health by reducing prematurity-related complications.

Referee #1 (Remarks for Author):

This is an interesting manuscript that addresses a pressing clinical issue. I have a few major questions and then some minor concerns.

Major Issues:

The experiment in depicted in Figure 4 shows that J2 does not affect the metabolism of P4. Moreover, in Figure 7C, neither J2 nor P4 affects levels of Akr1C18, an enzyme

involved in P4 metabolism. Why then does the combination of J2 and P4 cause a significant drop in *Akr1C18*?

Response: We thank Reviewer for an insightful question. Our results suggest that three key mechanisms are involved in stress-induced PTB by inhibiting progesterone receptor (PR) action (summarized in **Schema Figure 1**):

1. Stress increases the levels of FKBP51, which binds to progesterone receptor (PR) and inhibits its transcriptional activity (Guzeloglu-Kayisli, Semerci et al., 2021, Schatz, Guzeloglu-Kayisli et al., 2015).

2. Elevated FKBP51 requires 3.2-times higher EC50 of P4 (Hubler, Denny et al., 2003), meaning higher concentrations of P4 are needed to effectively activate PR to maintain pregnancy.

3. Stress also elevates levels of AKR1C18, an enzyme that metabolizes endogenous P4, reducing P4 bioavailability and, consequently, impairing its ability to activate PR.

While 15dPGJ2 alone does not inhibit the stress-induced increase in AKR1C18 on E18.25, it significantly reduces the stress-induced levels of FKBP51, which allows endogenous P4 to activate PR more efficiently. However, endogenous P4 is quickly metabolized by stress-induced AKR1C18, limiting P4 availability for PR activation. In contrast, when 15dPGJ2 and exogenous P4 are co-administered, FKBP51 levels are reduced, and P4 concentrations are elevated, enhancing PR activation more effectively despite the metabolic effects of AKR1C18 on endogenous P4. These dual actions provide an additive effect on PR activation, leading to further reductions in AKR1C18 levels and prolonging pregnancy more than either compound alone.

This hypothesis is also supported by additional findings: for instance, the administration of R5020, a progestin that is resistant to AKR1C18-mediated metabolism, prolongs gestation by inhibiting both stress-induced *Fkbp5* and *Akr1c18* (Fig. 7A and 7C, respectively), even when serum P4 levels are low (Fig. 6A). However, no additional benefit is observed when 15dPGJ2 is combined with R5020, compared to R5020 alone. Note that R5020-injected mice exhibited prolonged gestation, along with occurrences of stillbirth and dystocia (Fig. EV1A, B).

Why would restraint stress not significantly increase corticosterone levels at E18.5, while it does at earlier points in gestation? Please clarify the hypothesis that restraint stress accelerates the loss of the corpus luteum without increasing corticosterone levels.

Response: Our previous study (Guzeloglu-Kayisli et al., 2021) demonstrated that in *Fkbp5* wild-type

mice (*Fkbp5*^{+/+}), restraint stress (ST) increases corticosterone levels between E16 to E18, which decreases to near control levels detected in non-stressed *Fkbp5*^{+/+} mice (NST) on E18.25 (see **Fig.3B** from (Guzeloglu-Kayisli et al., 2021)).

In the current experiment (**Fig. 6B**), we measured corticosterone levels on E18.25, which means that by this time, corticosterone levels started to decline. This could explain why we did not observe significantly higher corticosterone levels in ST mice compared to NST mice at E18.25.

Regarding the hypothesis that restraint stress accelerates corpus luteum loss without increasing corticosterone levels, our previous study (Guzeloglu-Kayisli et al., 2021), found similar corpus luteal size and appearance between non-stressed (NST, control) and stressed (ST) mice. Histomorphometry analysis revealed no difference in luteal regression, and steroidogenic gene expression levels of *Cyp11a1*, (which encodes the enzyme converting cholesterol to pregnenolone, and *Hsd3b2* (converts pregnenolone to progesterone), were comparable between NST and ST conditions in ovaries collected at gestational day E17.25 and E18.25. These findings indicate that luteal regression in our stress model is independent of the corticosterone response (see **Suppl Fig.S5A, B**, from (Guzeloglu-Kayisli et al., 2021)).

There are many publications now showing how various pharmacotherapeutic approaches delay or prevent preterm birth in mice, e.g., agents that block IL-1 β , compounds that inhibit NF- κ B (our work) or antioxidants to reduce oxidative stress. Yet, as you mention in this manuscript, there is currently no FDA approved drug therapy to prevent preterm birth. What is different about the approach described in this manuscript that makes it more translatable and more likely to have a clinical impact than other drug therapies that have been explored?

Response: Approaches targeting IL-1 β or NF- κ B are particularly effective for preventing infection or inflammation-associated PTBs. These pathways play key roles in the inflammatory response, which is a significant contributor to infection-driven PTBs. By inhibiting, these strategies aim to reduce the inflammatory mediators that trigger PTB, thereby mitigating the risk of infection-induced PTBs. However, such interventions pose a risk to maternal and fetal health from severe chorioamnionitis and sepsis. Moreover, blockade of IL-1 β and/or NF- κ B signaling will be ineffective for stress-associated and idiopathic PTBs, which are not mediated by IL-1 β or NF- κ B signaling. As seen in **Fig. 7D and E**, maternal stress administration does not affect uterine *Ilf* and *Ptgs* levels at E18.25 in stress mice compared to control (NST). This highlights the need for alternative therapeutic strategies targeting different mechanisms.

Moreover, our unpublished data demonstrates that 15dPGJ2 significantly reduces the IL-1 β mediated increase in expression of *IL1B* and *PTGS2* mRNA levels in stromal cells, indicating a potential inhibitory effect on inflammatory pathways in these cells. However, while preventing PTB in a woman with chorioamnionitis is not our goal due to the aforementioned attendant maternal and fetal risks, nevertheless, given 15dPGJ2's strong anti-inflammatory properties (Chigusa, Kishore et al., 2016, Pirianov, Waddington et al., 2009) mediated by inhibition of NF- κ B signaling, key mediator of infection-associated labor (Forman, Tontonoz et al., 1995, Lindstrom & Bennett, 2005, Scher & Pillinger, 2005, Sykes, MacIntyre et al., 2014) and data presented in **Fig.3A-F** and unpublished data), it could also be beneficial in reducing early stage inflammation in infection-related PTB in settings where the mother and fetus are not at imminent peril. In addition, by preventing pre-parturition associated anatomic and pathophysiological changes in the lower genital tract, 15dPGJ2 could prevent subsequent ascending genital tract infection.

The brief statements "Given 15dPGJ2's potent anti-inflammatory properties, it may also be beneficial in reducing early-stage inflammation in infection-related PTB, particularly in situations where the mother and fetus are not at immediate risk." are now included in discussion of the revised manuscript (**Page 15**).

Minor Issues:

The figures appear stretched. Please provide better quality figures.

Response: All figures are provided with better quality as suggested by the reviewer.

Indicate statistical significance in Figure 1B.

Response: Statistical significance is now included in **Fig. 1B**.

Figure 2 does not indicate which panels are B and C.

Response: Fig. 2B and C are now indicated in the figure panels.

DSC stands for decidualized stromal cells. Therefore, is it redundant to say "decidualized DSC," (bottom of page 5)? Also, once the abbreviation is established, "decidualized stromal cells" should not be written out again (middle of page 7).

Response: You are correct that "decidualized DSC" is redundant, as DSC stands for decidualized stromal cells. To clarify, we have revised the statement from "decidualized DSC" to "*in vitro* decidualized DSC" to better reflect the condition of the cells. Additionally, we have changed "decidualized stromal cells" to "DSCs" on page 7, following standard abbreviation usage after the term is initially defined. These changes ensure consistency and avoid redundancy in the manuscript.

There is a critical typo in the first full paragraph on Page 15! (I believe 15dPG2 should be 15dPJ2.)

Response: We now corrected "15dPG2" as "15dPGJ2"

Centrifugation speeds should be expressed as how many g, not rpm.

Response: Centrifugation speeds are now expressed as "g" in the revised manuscript (**Page 19**).

There is no reference to Figure 8 in the text. Also, Figure 8 is out of focus in the copy provided.

Response: The Figure 8 is generated using Biorender. Now, copyright information is included in the figure legend.

Responses to Comments by Referee #2 (Comments on Novelty/Model System for Author):

Need clarity on why term decidual cultures were used to evaluate the impact on preterm birth

Response: We aimed to investigate how 15dPGJ2 modulates the expression of key genes and pathways involved in preterm birth (PTB). Term decidual cells are known to express critical receptors and signaling molecules involved in pregnancy maintenance and the onset of labor, making them a useful model for exploring mechanisms relevant to PTB. In our experiments, we also established several DSC cultures (n=2, 35 weeks) derived from preterm labor cases. Here again, 15dPGJ2 exhibited similar inhibitory effects on *FKBP5*, *IL1B* and *OXTR* levels as observed in term DSCs.

Referee #2 (Remarks for Author):

Guzeloglu-Kayisli and colleagues evaluate the role of 15dPGJ2 in mediating stress-induced preterm birth via the FKBP51-progesterone receptor. Guzeloglu-Kayisli and colleagues build on their prior work and have performed extensive studies on FKBP51 and the use of 15dPGJ2 to reduce preterm birth. Their work is elegant and culminates in a conceptual figure for a novel method to impact preterm birth. They demonstrate that 15dPGJ2/P4 co treatment reduces both FKBP51 and AKR1C18 levels, repressing labor-inducing genes.

The manuscript may benefit from addressing these items: The underlying biologic premise is linking maternal stress-related disorders to iatrogenic preterm birth and exploring the role of stress-mediated proteins and receptors in the pathway for preterm birth prevention. Maternal stress related disorders are listed as depression and fetal stress-related conditions including abnormal placentation. This is a broad categorization. Iatrogenic preterm birth likely has multiple etiologies including infectious, inflammatory, stress-related, genetic, etc.

Response: To further clarify, we provide a brief schematic of PTB classification based on timing, onset, and underlying cause (see **Table 1**). **Iatrogenic** (medically indicated) PTBs results from maternal or fetal health concerns, such as preeclampsia, fetal growth restriction, severe maternal cardiac or autoimmune diseases, requiring early delivery to ensure the safety of the mother or baby. This leads to the decision to induce labor or perform a cesarean delivery prematurely.

Timing	Preterm	Extreme Before 28 weeks
		Very Between 28 and 31 weeks
		Moderate to Late Between 32 and 36 weeks
Onset	< 37 Weeks	Spontaneous Labor begins naturally before 37 weeks
		Medically Indicated Labor induced/cesarian section due to maternal or fetal health concerns
Cause	Idiopathic Cause is unknown	
	Medical Health conditions that necessitate early delivery for the safety & wellbeing of mother or baby	

In contrast, **spontaneous PTBs**, which constitute 40–50% of all PTBs, are primarily caused by infection, inflammation, abruption, genetic factors or an unknown (idiopathic) mechanism (Goldenberg, Culhane et al., 2008, Lockwood, 2015, Savitz, Blackmore et al., 1991).

Our study specifically focuses on **idiopathic spontaneous PTB**, which is predominantly linked to stress-related disorders. As such, our study excludes PTBs induced by infection, inflammation, as well as iatrogenic PTBs.

It is not clear that direct correlation of these findings truly impacts this broad category of preterm birth.

Response: As mentioned in our previous response, our model specifically targets idiopathic spontaneous PTB, which is distinct from spontaneous PTB caused by infection or abruption. Additionally, while treatments for infection- and abruption-related preterm births, such as those proposed by several researchers, may be effective, they might not be clinically feasible due to concerns about maternal and fetal morbidity and mortality caused by complications of infection or abruption. Our focus on idiopathic preterm birth allows us to explore a more targeted approach that could be applicable to a broader range of cases within this category. For example, by preventing progressive cervical change, 15dPGJ2 could impede ascending genital tract infections or placental disruption causing abruption.

We patented use of 15dPGJ2 (Patent# US2022/0088032 (2022) "*Compositions and Uses Thereof for Treatment of Idiopathic Preterm Birth*") as a therapy for preventing of idiopathic PTB and have an agreement with Dare Bioscience Inc. for potential clinical translation. The findings presented in the manuscript are part of an ongoing research effort supported by an NIH STTR Phase I grant. Our successful Phase I results have laid a solid foundation for ongoing development of 15dPGJ2 + progesterone (P4), which we are formulating as a vaginal insert. In our proposed Phase II studies, we will focus on refining the duration and dosage of 15dPGJ2+P4 and defining its pharmacokinetic, safety, and toxicity profile in both pregnant and non-pregnant animal models for pre-clinical trials. After completing Phase II milestones, we plan to present our findings to the FDA during a pre-Investigational New Drug (IND) meeting for guidance on remaining pre-IND studies. We will be pursuing an initial indication for women with a singleton pregnancy and previous PTB or with a short cervix prior to 24 weeks without evidence of infection or abruption, groups at high risk for spontaneous idiopathic PTB (Frey, McLaughlin et al., 2022). Therefore, this approach builds on existing research and could provide a promising alternative to current strategies for preventing PTB. By preventing stress-induced PTB and extending gestation, this product has the potential to significantly improve neonatal health by reducing prematurity-related complications.

The withdrawal of Makena is highlighted as a major rationale for this investigation. This work builds on prior work by this group exploring the role of FKBP5 in preterm birth. The withdrawal of Makena is a distractor. Even were it still approved, other novel approaches to impact preterm birth are needed.

The evaluations of 15dPGJ2 - with fatty acid derivatives inhibit glucocorticoid induced FKBP51 levels, on levels of labor initiating molecules, and others all use term DSC cultures - why were they not evaluated in preterm DSC cultures?

Response: We previously compared first trimester decidual stromal cells (FTDC) and term decidual stromal cells (TDC) in culture and found no significant differences in endogenous *PGR* or *FKBP5* levels between the two cell types. Additionally, the medroxyprogesterone acetate (MPA, a progestin/glucocorticoid mix)-induced increases in *FKBP5* levels that were similar in

both groups. This suggests that, under *in vitro* conditions, the response is consistent and does not vary with gestational age.

Figure 1. *PGR* and *FKBP5* levels in FTDC and TDC cultures treated with 10^{-8} M Estradiol (E)± 10^{-7} M Medroxyprogesterone (MPA) by qPCR. Bars represent mean± SEM, n=3, * $P<0.05$ vs E.

In addition, as noted in the response above, we have generated two “preterm” DSCs cultures inhibited *FKBP5*, *IL1B*, and *OXTR* levels in these cultures, identical to the inhibitory effects observed in term DSC derived from women with preterm births at 35 weeks of gestation and observed that 15dPGJ2 cultures.

As pregnancy advances there are remarkable changes in the maternal, fetal, placental, and uterine physiology and metabolism. The decidua changes and has a different composition of immune and inflammatory markers. Why not perform these studies in preterm decidua?

Response: We agreed with Reviewer’s comment that decidua primarily consists of DSCs, leukocytes and trophoblasts, but progesterone receptors (**PRs**) are expressed solely in DSCs (Lockwood, Stocco et al., 2010, Merlino, Welsh et al., 2009, Suryawanshi, Morozov et al., 2018, Vento-Tormo, Efremova et al., 2018). In our previous work (Guzeloglu-Kayisli et al., 2021), we did evaluate preterm decidual tissues from idiopathic PTBs vs. gestational age (**GA**)-matched controls obtained from preeclampsia and placenta accreta. We found that *FKBP5* levels are significantly elevated in idiopathic PTB. Using immunohistochemistry, we confirmed the increased nuclear expression of FKBP51 in decidual cells, and through *in situ* proximity ligation assay, we demonstrated significantly elevated nuclear FKBP51-PR interactions in idiopathic PTBs compared to gestational age-matched controls, including C-sections from pregnancies complicated by preeclampsia or placenta accreta. This prior analysis of preterm decidua supports the relevance of our current findings.

The mouse model utilized restraint stress between E8 and E18 in three sessions for one hour - does this mean three sessions each day? Or three sessions of one hour at some timepoint between E8 and E18? Did they have access to food and water while restrained?

Response: Restraint stress was administered daily from E8 to E18, with three separate stress sessions per day, totaling 3 hours of stress exposure daily. The stress schedule was variable to prevent habituation to the restraint stress. During the stress sessions, there was no access to food or water. However, as previously reported, no significant differences in food consumption or water intake were observed throughout gestation between the nonstress and stress groups (Guzeloglu-Kayisli et al., 2021) by measuring the amount of food and water consumed weekly.

How were the sample sizes determined? I do not see that in the statistical analysis section.

Response: This is a study supported by NIH and at the time of application we performed samples size calculation. We now include this information in the revised manuscript as suggested by the reviewer (**Page 22**).

Sample size calculation in *in vitro* studies based on our qPCR assessment of *FKBP5* expression in TDC cultures, in which we observed that *FKBP5* mRNA levels decrease by ~50% by 15PGJ2+Dex vs. Dex alone treatment (standard deviation: ~20%; $P < 0.01$). We aim to detect a similar difference in *FKBP5* mRNA and protein levels among 6 different treatment groups using One-Way ANOVA with *post-hoc* Tukey tests. Thus, to achieve 90% power with an alpha (type I error) of 0.01, and assuming the same standard deviation, we require 5 biological repeats of TDC cultures, isolated from separate individuals per treatment group to achieve statistical significance.

Sample size calculation in *mice studies* is based on our previous publication (Guzeloglu-Kayisli et al., 2021) in which C57BL/6 mice are reported to have a gestational length of 19.2 ± 0.6 days (mean \pm SD) and 18.7 ± 0.1 days under restraint stress. In *preliminary results*, 15PGJ2 administration to restrained mice significantly improved gestational length (mean \pm SD 19.27 ± 0.52 days, $P < 0.05$) and permitted 62.5% of restrained mice to deliver on day 19.0 or thereafter. We aim to detect a similar difference among 5 different groups using One-Way ANOVA with *post-hoc* Tukey tests. Thus, to achieve 80% power with an alpha (type I error) of 0.05, and assuming the same standard deviation detected in 15PGJ2 administration in restrained mice, we require 11 mice per treatment group to achieve statistical significance.

References

- Chigusa Y, Kishore AH, Mogami H, Word RA (2016) Nrf2 Activation Inhibits Effects of Thrombin in Human Amnion Cells and Thrombin-Induced Preterm Birth in Mice. *J Clin Endocrinol Metab* 101: 2612-21
- Forman BM, Tontonoz P, Chen J, Brun RP, Spiegelman BM, Evans RM (1995) 15-Deoxy-delta 12, 14-prostaglandin J2 is a ligand for the adipocyte determination factor PPAR gamma. *Cell* 83: 803-12
- Frey HA, McLaughlin EM, Hade EM, Finneran MM, Rood KM, Shellhaas C, Landon MB (2022) Obstetric History and Risk of Short Cervix in Women with a Prior Preterm Birth. *Am J Perinatol* 39: 759-765
- Goldenberg RL, Culhane JF, Iams JD, Romero R (2008) Epidemiology and causes of preterm birth. *Lancet* 371: 75-84
- Guzeloglu-Kayisli O, Semerci N, Guo X, Larsen K, Ozmen A, Arlier S, Mutluay D, Nwabuobi C, Sipe B, Buhimschi I, Buhimschi C, Schatz F, Kayisli UA, Lockwood CJ (2021) Decidual cell FKBP51-progesterone receptor binding mediates maternal stress-induced preterm birth. *Proc Natl Acad Sci U S A* 118
- Hubler TR, Denny WB, Valentine DL, Cheung-Flynn J, Smith DF, Scammell JG (2003) The FK506-binding immunophilin FKBP51 is transcriptionally regulated by progestin and attenuates progestin responsiveness. *Endocrinology* 144: 2380-7
- Lindstrom TM, Bennett PR (2005) 15-Deoxy-delta12,14-prostaglandin j2 inhibits interleukin-1beta-induced nuclear factor-kappaB in human amnion and myometrial cells: mechanisms and implications. *J Clin Endocrinol Metab* 90: 3534-43
- Lockwood CJ (2015) Risk factors for preterm birth and new approaches to its early diagnosis. *J Perinat Med* 43: 499-501

Lockwood CJ, Stocco C, Murk W, Kayisli UA, Funai EF, Schatz F (2010) Human labor is associated with reduced decidual cell expression of progesterone, but not glucocorticoid, receptors. *J Clin Endocrinol Metab* 95: 2271-5

Merlino A, Welsh T, Erdonmez T, Madsen G, Zakar T, Smith R, Mercer B, Mesiano S (2009) Nuclear progesterone receptor expression in the human fetal membranes and decidua at term before and after labor. *Reprod Sci* 16: 357-63

Pirianov G, Waddington SN, Lindstrom TM, Terzidou V, Mehmet H, Bennett PR (2009) The cyclopentenone 15-deoxy-delta 12,14-prostaglandin J(2) delays lipopolysaccharide-induced preterm delivery and reduces mortality in the newborn mouse. *Endocrinology* 150: 699-706

Savitz DA, Blackmore CA, Thorp JM (1991) Epidemiologic characteristics of preterm delivery: etiologic heterogeneity. *Am J Obstet Gynecol* 164: 467-71

Schatz F, Guzeloglu-Kayisli O, Basar M, Buchwalder LF, Ocak N, Guzel E, Guller S, Semerci N, Kayisli UA, Lockwood CJ (2015) Enhanced Human Decidual Cell-Expressed FKBP51 May Promote Labor-Related Functional Progesterone Withdrawal. *Am J Pathol* 185: 2402-11

Scher JU, Pillinger MH (2005) 15d-PGJ2: the anti-inflammatory prostaglandin? *Clin Immunol* 114: 100-9

Suryawanshi H, Morozov P, Straus A, Sahasrabudhe N, Max KEA, Garzia A, Kustagi M, Tuschl T, Williams Z (2018) A single-cell survey of the human first-trimester placenta and decidua. *Sci Adv* 4: eaau4788

Sykes L, MacIntyre DA, Teoh TG, Bennett PR (2014) Anti-inflammatory prostaglandins for the prevention of preterm labour. *Reproduction* 148: R29-40

Vento-Tormo R, Efremova M, Botting RA, Turco MY, Vento-Tormo M, Meyer KB, Park JE, Stephenson E, Polanski K, Goncalves A, Gardner L, Holmqvist S, Henriksson J, Zou A, Sharkey AM, Millar B, Innes B, Wood L, Wilbrey-Clark A, Payne RP et al. (2018) Single-cell reconstruction of the early maternal-fetal interface in humans. *Nature* 563: 347-353

19th Feb 2025

Dear Dr. Guzeloglu Kayisli,

Thank you for the submission of your revised manuscript to EMBO Molecular Medicine. I am pleased to inform you that we will be able to accept your manuscript pending the following final amendments:

- 1) Title: Please consider revising the title, e.g. Targeting FKBP51 prevents stress-induced preterm birth.
- 2) Abstract: I have gone through your text and revised it (see below). Please review it, amend as you see fit:

Preterm birth (PTB) is a leading cause of perinatal morbidity and mortality, with maternal stress-related disorders, such as depression and anxiety, linked to idiopathic PTB (iPTB). At the maternal-fetal interface, decidualized stromal cells (DSCs) exclusively express the progesterone receptor (PR) and play pivotal roles in maintaining pregnancy and initiating labor. DSCs also express FKBP51, a protein that binds to and inhibits glucocorticoid and PR receptors and is associated with stress-related diseases. We previously found that iPTB specimens exhibit increased FKBP51 levels and enhanced FKBP51-PR interactions in DSC nuclei. Additionally, we demonstrated that *Fkbp5*-deficient mice have prolonged gestation and are resistant to stress-induced PTB, suggesting that FKBP51 contributes to iPTB pathogenesis. Since no FDA-approved therapy exists for PTB, we hypothesized that inhibiting FKBP51 could prevent iPTB. Our current results show that the endogenous prostaglandin D2 derivative 15dPGJ2 reduces FKBP51 levels and FKBP51-PR interactions in cultured cells. Maternal stress increases uterine expression of *Fkbp5*, *Oxtr*, and *Akr1c18*, leading to shortened gestation. However, treatment with 15dPGJ2 lowers uterine *Fkbp51*, *Oxtr*, and *Ptgs2* levels and prevents stress-induced PTB. Notably, co-treatment with 15dPGJ2 and either P4 or R5020 produced the most significant effects, highlighting the potential of 15dPGJ2 alone or in combination with progestins as a promising therapeutic strategy to prevent PTB.

- 3) Figures: Please rationalize the number of figures. We note that many figures have only 2-3 panels, e.g. Fig. 1 and 2 could be merged etc. Please check "Author Guidelines" for more information:

<https://www.embopress.org/page/journal/17574684/authorguide#figureformat>

<https://www.embopress.org/page/journal/17574684/authorguide#expandedview>

- 4) In the main manuscript file, please do the following:

- Please address all comments suggested by our data editors listed below:

- o Figure legends:

1. Please note that figure 8 is not provided in the manuscript, however legend for the same is missing. Kindly rectify the same.
 2. Please note that the exact p values are not provided in the legends of figures 1A, 2A, 3A, B, C, E; 4A, B; 5E, F; 6A, 7A-D; EV1 A.
 3. Please note that scale bar and its definition are missing for figure EV3
- Place keywords after the abstract.
 - Remove data not shown (p.9).
 - Please remove callouts for Fig. 8 that is now synopsis image.
 - Remove Table EV1 and its legend. The legend should be added to the Table EV1 excel file.
 - In Methods, provide the antibody dilutions that were used for each antibody.
 - In Methods, provide primer sequences used for the RT-PCR reactions.
 - Indicate in legends exact n and exact p values, not a range, along with the statistical test used. To keep the figures "clear" some authors found providing an Appendix table Sx with all exact p-values preferable. You are welcome to do this if you want to.
 - Author contributions: Please remove it from the manuscript and specify author contributions in our submission system. CRediT has replaced the traditional author contributions section because it offers a systematic machine-readable author contributions format that allows for more effective research assessment. You are encouraged to use the free text boxes beneath each contributing author's name to add specific details on the author's contribution. More information is available in our guide to authors:

<https://www.embopress.org/page/journal/17574684/authorguide#authorshippinguidelines>

- Data availability: Please replace the current text with the sentence: "This study includes no data deposited in external repositories."

- Correct the reference citation in the reference list. Citations should be listed in alphabetical order. Where there are more than 10 authors on a paper, 10 will be listed, followed by "et al.". Remove DOIs and PMID/PMCID numbers. Please check "Author Guidelines" for more information.

<https://www.embopress.org/page/journal/17574684/authorguide#referencesformat>

- 5) Reagent Table: Please mark with an asterisk at the appropriate place to refer to the reference on the end of the file.

- 6) Synopsis:

- Synopsis image: Please resize the image to 550 px-wide x (300-600)-px high and upload it as a high-resolution jpeg file
- Synopsis text: Please remove it from the main manuscript file and upload it as a separate .doc file.

- 7) Source data: Please upload source data as one folder per figure.

- 8) As part of the EMBO Publications transparent editorial process initiative (see our Editorial at

<http://embomolmed.embopress.org/content/2/9/329>), EMBO Molecular Medicine will publish online a Review Process File (RPF)

to accompany accepted manuscripts. This file will be published in conjunction with your paper and will include the anonymous referee reports, your point-by-point response and all pertinent correspondence relating to the manuscript. Let us know whether you agree with the publication of the RPF and as here, if you want to remove or not any figures from it prior to publication. Please note that the Authors checklist will be published at the end of the RPF.

9) Please provide a point-by-point letter INCLUDING my comments as well as the reviewer's reports and your detailed responses (as Word file).

I look forward to reading a new revised version of your manuscript as soon as possible.

Yours sincerely,

Zeljko Durdevic

Zeljko Durdevic
Senior Editor
EMBO Molecular Medicine

*** Instructions to submit your revised manuscript ***

- 1) a .docx formatted version of the manuscript text (including Figure legends and tables)
- 2) Separate figure files*
- 3) supplemental information as Expanded View and/or Appendix. Please carefully check the authors guidelines for formatting Expanded view and Appendix figures and tables at <https://www.embopress.org/page/journal/17574684/authorguide#expandedview>
- 4) a letter INCLUDING the reviewer's reports and your detailed responses to their comments (as Word file).
- 5) The paper explained: EMBO Molecular Medicine articles are accompanied by a summary of the articles to emphasize the major findings in the paper and their medical implications for the non-specialist reader. Please provide a draft summary of your article highlighting
 - the medical issue you are addressing,
 - the results obtained and
 - their clinical impact.This may be edited to ensure that readers understand the significance and context of the research. Please refer to any of our published articles for an example.
- 6) Author contributions: the contribution of every author must be detailed in a separate section.
- 7) EMBO Molecular Medicine now requires a complete author checklist (<https://www.embopress.org/page/journal/17574684/authorguide>) to be submitted with all revised manuscripts. Please use the checklist as guideline for the sort of information we need WITHIN the manuscript. The checklist should only be filled with page

numbers were the information can be found. This is particularly important for animal reporting, antibody dilutions (missing) and exact values and n that should be indicted instead of a range.

8) Every published paper now includes a 'Synopsis' to further enhance discoverability. Synopses are displayed on the journal webpage and are freely accessible to all readers. They include a short stand first (maximum of 300 characters, including space) as well as 2-5 one sentence bullet points that summarise the paper. Please write the bullet points to summarise the key NEW findings. They should be designed to be complementary to the abstract - i.e. not repeat the same text. We encourage inclusion of key acronyms and quantitative information (maximum of 30 words / bullet point). Please use the passive voice. Please attach these in a separate file or send them by email, we will incorporate them accordingly.

You are also welcome to suggest a striking image or visual abstract to illustrate your article. If you do please provide a jpeg file 550 px-wide x 300-600px high.

9) A Conflict of Interest statement should be provided in the main text

10) Please note that we now mandate that all corresponding authors list an ORCID digital identifier. This takes <90 seconds to complete. We encourage all authors to supply an ORCID identifier, which will be linked to their name for unambiguous name identification.

Currently, our records indicate that the ORCID for your account is 0000-0001-5448-917X.

Please click the link below to modify this ORCID:
Link Not Available

11) Include a Reagents and Tools Table as part of the Methods section, which can be downloaded from our author guidelines (<https://www.embopress.org/page/journal/17574684/authorguide#structuredmethods>)

Graphs 800-1,200 DPI
Photos 400-800 DPI
Colour (only CMYK) 300-400 DPI"

*Additional important information regarding figures and illustrations can be found at
<https://bit.ly/EMBOPressFigurePreparationGuideline>. See also figure legend preparation guidelines:
<https://www.embopress.org/page/journal/17574684/authorguide#figureformat>

***** Reviewer's comments *****

Referee #1 (Comments on Novelty/Model System for Author):

The model system is considered adequate in the field.

Referee #1 (Remarks for Author):

I believe that all my concerns have been addressed and the manuscript is now suitable for publication.

Referee #2 (Comments on Novelty/Model System for Author):

addressed all points of reviews

Referee #2 (Remarks for Author):

concerns from reviews have been addressed

February 20, 2025

Dear Dr. Durdevic:

We would like to express our sincere gratitude to you and the reviewers for the prompt and thorough review of our manuscript (EMM-2024-21138-V2) titled “**15dPGJ2 inhibits maternal stress-induced preterm birth by reducing FKBP51-progesterone receptor interaction**” We greatly appreciate the valuable feedback provided, and no additional comments have been requested by the reviewers. In response to your feedback, we have carefully revised the manuscript. All changes made in the revised version are highlighted in yellow for easy reference.

1) Title: Please consider revising the title, e.g. Targeting FKBP51 prevents stress-induced preterm birth.

Response: The title is now revised as suggested.

2) Abstract: I have gone through your text and revised it (see below). Please review it, amend as you see fit:

“Preterm birth (PTB) is a leading cause of perinatal morbidity and mortality, with maternal stress-related disorders, such as depression and anxiety, linked to idiopathic PTB (IPTB). At the maternal-fetal interface, decidualized stromal cells (DSCs) exclusively express the progesterone receptor (PR) and play pivotal roles in maintaining pregnancy and initiating labor. DSCs also express FKBP51, a protein that binds to and inhibits glucocorticoid and PR receptors and is associated with stress-related diseases. We previously found that IPTB specimens exhibit increased FKBP51 levels and enhanced FKBP51-PR interactions in DSC nuclei. Additionally, we demonstrated that *Fkbp5*-deficient mice have prolonged gestation and are resistant to stress-induced PTB, suggesting that FKBP51 contributes to IPTB pathogenesis. Since no FDA-approved therapy exists for PTB, we hypothesized that inhibiting FKBP51 could prevent IPTB. Our current results show that the endogenous prostaglandin D2 derivative 15dPGJ2 reduces FKBP51 levels and FKBP51-PR interactions in cultured cells. Maternal stress increases uterine expression of *Fkbp5*, *Oxtr*, and *Akr1c18*, leading to shortened gestation. However, treatment with 15dPGJ2 lowers uterine *Fkbp5*, *Oxtr*, and *Ptgs2* levels and prevents stress-induced PTB. Notably, co-treatment with 15dPGJ2 and either P4 or R5020 produced the most significant effects, highlighting the potential of 15dPGJ2 alone or in combination with progestins as a promising therapeutic strategy to prevent PTB.”

Response: The revised abstract is accepted without any change.

3) Figures: Please rationalize the number of figures. We note that many figures have only 2-3 panels, e.g. Fig. 1 and 2 could be merged etc. Please check "Author Guidelines" for more information:

<https://www.embopress.org/page/journal/17574684/authorguide#figureformat>

<https://www.embopress.org/page/journal/17574684/authorguide#expandedview>

Response: To streamline the number of figures, we have combined Figures 1 and 2, which are now presented as **Figure 1A-E**. All corresponding changes have been made throughout the manuscript.

4) In the main manuscript file, please do the following:

- Please address all comments suggested by our data editors listed below:

Figure legends:

1. **Please note** that figure 8 is not provided in the manuscript, however legend for the same is missing. Kindly rectify the same.

Response: The Figure 8 (now Figure 7) and its legend are provided.

2. Please note that the exact p values are not provided in the legends of figures 1A, 2A, 3A, B, C, E; 4A, B; 5E, F; 6A, 7A-D; EV1 A.

Response: The exact p-values have already been provided where they were calculated through statistical analysis. However, in cases where the statistical analysis yielded p-values less than 0.05, 0.01, or 0.001, we have indicated these values in the figure legends instead of providing the exact numbers.

3. **Please note** that scale bar and its definition are missing for figure EV3

Response: The scale bar is included in **Figure EV3**, with its definition provided in the legend.

- **Place** keywords after the abstract.

Response: Keywords have been moved to after the abstract.

- **Remove** data not shown (p.9).

Response: "*data not shown*" is removed on P9.

- **Please remove** callouts for Fig. 8 that is now synopsis image.

Response: The callouts in the figure are now removed (now Figure 7 and synopsis image).

- **Remove** Table EV1 and its legend. The legend should be added to the Table EV1 excel file.

Response: **Table EV1** and its legend have been removed from the manuscript and uploaded as an Excel file.

- **In Methods**, provide the antibody dilutions that were used for each antibody.

Response: Primary and secondary antibody dilutions are provided on P19.

- **In Methods**, provide primer sequences used for the RT-PCR reactions.

Response: All primers were purchased from Thermo-Fisher as TaqMan gene expression assay. Thus, ID number for each primer is provided in Reagent and Tools Table.

- **Indicate in legends** exact n and exact p values, not a range, along with the statistical test used. To keep the figures "clear" some authors found providing an Appendix table Sx with all exact p-values preferable. You are welcome to do this if you want to.

Response: Each figure includes the sample size ("n") and p-values, along with the statistical test used. Exact p-values are provided where calculated through statistical analysis. However, when the p-values were less than 0.05, 0.01, or 0.001, we indicated these ranges in the figure legends instead of providing the exact numbers.

- **Author contributions:** Please remove it from the manuscript and specify author contributions in our submission system. CRedit has replaced the traditional author contributions section because it offers a systematic machine-readable author contributions format that allows for more effective research

assessment. You are encouraged to use the free text boxes beneath each contributing author's name to add specific details on the author's contribution. More information is available in our guide to authors:

<https://www.embopress.org/page/journal/17574684/authorguide#authorshipguidelines>

Response: The Author contribution section has been removed and replaced with the CRediT system.

- **Data availability:** Please replace the current text with the sentence: "This study includes no data deposited in external repositories."

Response: The sentence has been replaced with the suggested sentence in the Data Availability section.

- Correct the reference citation in the reference list. Citations should be listed in alphabetical order. Where there are more than 10 authors on a paper, 10 will be listed, followed by "et al.". Remove DOIs and PMID/PMCID numbers. Please check "Author Guidelines" for more information.
<https://www.embopress.org/page/journal/17574684/authorguide#referencesformat>

Response: The references have been revised to match the Journal's style.

5) Reagent Table: Please mark with an asterisk at the appropriate place to refer to the reference on the end of the file.

Response: The reference has been removed and marked with a "*" in the Reagent and Tool file, as suggested.

6) Synopsis: - Synopsis image: Please resize the image to 550 px-wide x (300-600)-px high and upload it as a high-resolution jpeg file

- **Synopsis text:** Please remove it from the main manuscript file and upload it as a separate .doc file.

Response: The synopsis text has been removed from the main manuscript and uploaded separately. The synopsis figure is provided as a high-resolution JPEG file (550 px x 300 px), as suggested.

7) Source data: Please upload source data as one folder per figure.

Response: Source data are uploaded as one folder per figure.

8) As part of the EMBO Publications transparent editorial process initiative (see our Editorial at <http://embomolmed.embopress.org/content/2/9/329>), EMBO Molecular Medicine will publish online a Review Process File (RPF) to accompany accepted manuscripts. This file will be published in conjunction with your paper and will include the anonymous referee reports, your point-by-point response and all pertinent correspondence relating to the manuscript. Let us know whether you agree with the publication of the RPF and as here, if you want to remove or not any figures from it prior to publication. Please note that the Authors checklist will be published at the end of the RPF.

Response: We agree with the publication of the RPF. The previously published figures have now been removed from the first RPF.

9) Please provide a point-by-point letter INCLUDING my comments as well as the reviewer's reports and your detailed responses (as Word file).

Response: Point-by point letter with editorial comments is included.

EMBO Molecular Medicine

*** Instructions to submit your revised manuscript ***

Response: We agree with the publication of the RPF.

To submit your manuscript, please follow this link:

<https://embomolmed.msubmit.net/cgi-bin/main.plex>

1) a **.docx** formatted version of the manuscript text (including Figure legends and tables)

Response: Docs formatted main manuscript with figures and table legends are uploaded.

2) **Separate** figure files*

Response: Each figure has been uploaded separately as a TIFF file with 800 dpi resolution. Figure panels are labeled with capital letters in Arial font. Each figure includes its own separate legend.

3) **supplemental** information as Expanded View and/or Appendix. Please carefully check the authors guidelines for formatting Expanded view and Appendix figures and tables at <https://www.embopress.org/page/journal/17574684/authorguide#expandedview>

Response: Supplemental figures are given as Expanded view (Figures EV1-3).

4) a **letter** INCLUDING the reviewer's reports and your detailed responses to their comments (as Word file).

Response: Cover letter and point-by point response files have been provided.

5) **The paper explained:** EMBO Molecular Medicine articles are accompanied by a summary of the articles to emphasize the major findings in the paper and their medical implications for the non-specialist reader. Please provide a draft summary of your article highlighting

This may be edited to ensure that readers understand the significance and context of the research.

Response: The paper explained, as shown below, was previously included in the manuscript.

“The paper explained

Problem

Preterm birth (PTB), defined as delivery before 37 weeks of gestation, affects 10.5% of all U.S. births, with a higher rate of 14-15% among Black women. The stress response co-chaperone FKBP51 binds to progesterone receptors (PRs), inhibiting PR-mediated transcriptional activity, which leads to functional

progesterone withdrawal and results in stress-associated and idiopathic PTB. This study investigates the efficacy of 15dPGJ2, alone or in combination with progesterone or synthetic progestin R5020, in preventing maternal stress-induced PTB and evaluates its potential as anti-PTB therapeutic.

Results

We found that 15dPGJ2 prevented maternal stress-induced PTB in mice by reducing levels of *Fkbp5* and labor-inducing mediators. 15dPGJ2 co-treatment with progestin, further extended gestation, suggesting 15dPGJ2 as a promising anti-PTB therapy.

Impact

Our findings demonstrate that 15dPGJ2 targets FKBP51, a key regulator of the stress response pathway, significantly inhibiting its levels in a safe and effective manner. This therapy has the potential to fundamentally change clinical practice and reduce PTBs. Inhibition of uterine FKBP51 activity could also have broad applicability in other causes of PTB. Furthermore, elevated FKBP51 levels are linked to stress-related psychiatric disorders, type 2 diabetes, and obesity, suggesting that targeting FKBP51 could offer therapeutic benefits beyond obstetrics.”

6) Author contributions: the contribution of every author must be detailed in a separate section.

Response: Author contributions are provided in the journal submission system.

7) EMBO Molecular Medicine now requires a complete author checklist (<https://www.embopress.org/page/journal/17574684/authorguide>) to be submitted with all revised manuscripts. Please use the checklist as guideline for the sort of information we need WITHIN the manuscript. The checklist should only be filled with page numbers where the information can be found. This is particularly important for animal reporting, antibody dilutions (missing) and exact values and n that should be indicated instead of a range.

Response: Revised Author Checklist is provided.

8) Every published paper now includes a 'Synopsis' to further enhance discoverability. Synopses are displayed on the journal webpage and are freely accessible to all readers. They include a short stand first (maximum of 300 characters, including space) as well as 2-5 one sentence bullet points that summarise the paper. Please write the bullet points to summarize the key NEW findings. They should be designed to be complementary to the abstract - i.e. not repeat the same text. We encourage inclusion of key acronyms and quantitative information (maximum of 30 words / bullet point). Please use the passive voice. Please attach these in a separate file or send them by email, we will incorporate them accordingly.

You are also welcome to suggest a striking image or visual abstract to illustrate your article. If you do please provide a jpeg file 550 px-wide x 300-600px high.

Response: The synopsis text has been removed from the main manuscript and uploaded separately. The synopsis figure is provided as a high-resolution JPEG file (550 px x 300 px), as suggested.

9) A Conflict of Interest statement should be provided in the main text.

Response: Conflict of interest statement “All authors declare that they have no further competing interests” is now added on P22.

10) Please note that we now mandate that all corresponding authors list an ORCID digital identifier. This takes <90 seconds to complete. We encourage all authors to supply an ORCID identifier, which will be linked to their name for unambiguous name identification.

Currently, our records indicate that the ORCID for your account is 0000-0001-5448-917X.

Please click the link below to modify this ORCID: <https://embomolmed.msubmit.net/cgi-bin/main.plex?el=A1E11BEtg4B3DIAf4Bh7B9ftdpWL5rvBJitnQVfz8HSNAY>

Response: The link is successfully linked to my account.

11) Include a Reagents and Tools Table as part of the Methods section, which can be downloaded from our [author guidelines](https://www.embopress.org/page/journal/17574684/authorguide#structuredmethods) (<https://www.embopress.org/page/journal/17574684/authorguide#structuredmethods>)

Response: Revised reagents and Tools Table is uploaded.

24th Feb 2025

Dear Dr. Guzeloglu Kayisli,

We are pleased to inform you that your manuscript is accepted for publication and is now being sent to our publisher to be included in the next available issue of EMBO Molecular Medicine.

Zeljko Durdevic
Senior Editor
EMBO Molecular Medicine
